# CRIME PREDICTION USING ADAPTIVE QUADTREES

## ABSTRACT

Urban crime prediction demands scalable methods for large, skewed spatio-temporal data. We introduce SMART-CARE, an adaptive quadtree-based hierarchical framework that dynamically partitions urban spaces and refines local predictors. Given $\mathcal{D} = \{(\mathbf{x}_i, t_i, c_i)\}_{i=1}^{N}$, SMART-CARE learns $f : (\mathbf{x}, t) \mapsto \hat{c}$ through: (i) variance-driven median splitting with adaptive capacity $T_{\max}$ and depth $L_{\max}$, (ii) periodic local re-tuning with leaf merging to prevent over-fragmentation, and (iii) parent→child knowledge transfer for model fine-tuning. Experiments on NYC and Chicago crime data show SMART-CARE outperforms uniform grids, static quadtrees, and standard baselines in accuracy and efficiency while enabling fine-grained localized forecasts.

## 1 INTRODUCTION

Urban crime forecasting remains challenging due to irregular spatio-temporal patterns and data skew in heterogeneous urban environments Zhang et al. (2015). Existing approaches struggle to capture complex dependencies in high-density regions, where skewed distributions obscure local trends and reduce prediction reliability Safat et al. (2021); Xiong et al. (2019). While recent machine learning methods have improved accuracy, they often fail to model hierarchical spatial scales-specific patterns essential for both city-wide trends and neighborhood-level variations Groff et al. (2019); Lu et al. (2021). This limitation underscores the need for adaptive frameworks that dynamically refine predictions across spatial granularities while maintaining scalability.

The challenges are particularly critical in high-density urban areas, where traditional methods struggle to distinguish recurring patterns from isolated events Mahmud et al. (2016). Data skew in these regions often leads models to overfit to dense hotspots while neglecting significant crime patterns in lower-density areas Tayebi et al. (2015); Kennedy et al. (2011). This spatial bias results in poor generalization across diverse urban geographies Braga et al. (2019); Lin et al. (2017), highlighting the need for methods that explicitly account for density variations and spatial variance.

Butt et al. (2021) Butt et al. (2021) demonstrate that density-based clustering like HDBSCAN often produces imbalanced partitions where dense regions overshadow sparse ones, degrading time-series methods that assume balanced inputs. Their finding that smaller, homogeneous regions yield better local forecasts motivates partitioning strategies that balance segment sizes while accounting for local variance—addressing both scalability and accuracy across diverse urban environments.

Recent work by Butt et al. (2024) Butt et al. (2024) explores BiLSTM-based transfer learning for crime forecasting, pretraining on source regions and fine-tuning on targets. While this approach learns transferable temporal patterns, it requires substantial pretraining data and compute, risks negative transfer across heterogeneous regions, and offers limited interpretability. These limitations highlight the need for more efficient, spatially-aware knowledge transfer mechanisms.

To address these limitations, we propose **SMART-CARE**, an adaptive quadtree framework that integrates variance-aware median splitting with hierarchical model refinement. Our approach dynamically adjusts spatial partitions while merging low-density nodes to prevent over-fragmentation. Key innovations include: (i) *feature-propagation*, where parent predictions inform child feature vectors, and (ii) *model-inheritance*, where child models are warm-started from parent parameters and fine-tuned locally. We instantiate SMART-CARE with both tree-based and neural architectures, demonstrating its flexibility. On NYPD and Chicago crime data (2008–2023), SMART-CARE achieves MAE reductions to 0.92 (average), 0.23 (best-year), and 0.13 (deepest layer), significantly outperforming grid-based, clustering, and transfer learning baselines Butt et al. (2021; 2024).

## 2 RELATED WORK

Crime prediction research enhances public safety by identifying hotspots using historical data Du & Ding (2023). While clustering and time-series models address data imbalance and spatial unevenness Butt et al. (2021), they often struggle with high spatial variance and density bias, causing overfitting in heterogeneous regions Mahmud et al. (2016). Furthermore, the absence of hierarchical refinement limits their scalability across diverse urban settings. These limitations underscore the need for adaptive, multi-scale systems that balance data and improve predictive accuracy.

**Data Imbalance:** Crime data often exhibits skewed distributions, with non-crime instances vastly outnumbering crime events, challenging model performance. Techniques like Bayesian classification Tang et al. (2016), SVMs for imbalanced tasks Tang et al. (2008), and synthetic oversampling methods such as ADASYN He et al. (2008) and SMOTE for regression Torgo et al. (2013) attempt to address imbalance but risk introducing artifacts that distort the underlying distribution. These approaches often fail to preserve spatial relationships, leading to inaccurate predictions in dense regions. This highlights the need for methods that balance geographical segmentation, ensuring localised accuracy while mitigating the effects of skewed data distributions in crime prediction.

**Hierarchical Data Structure:** Hierarchical data structures are widely used for spatial tasks, with recent advancements enhancing crime prediction models. DeepCrime Huang et al. (2018) leverages a region-category encoder, hierarchical recurrent layers, and attention mechanisms to capture temporal dependencies, achieving notable accuracy. However, its reliance on Gated Recurrent Units (GRU) incurs high computational overhead for large datasets. Our hierarchical quadtree framework mitigates this by offering scalable, spatially adaptive segmentation, reducing computational demands while preserving fine-grained prediction accuracy, making it suitable for large-scale crime prediction across diverse spatial scales.

Zhou et. al., 2019 Zhou et al. (2019) proposed DENSS, a semi-supervised learning framework using Density Peak Clustering (DPC) with an R-tree index to ensure uniform local densities. While R-trees employ minimum bounding rectangles (MBRs) for hierarchical spatial organisation, overlapping MBRs increase computational complexity. In contrast, our framework dynamically segments crime regions into balanced quadrants, reducing data skewness and enhancing scalability. This efficient hierarchical modelling approach enables precise crime predictions across diverse spatial scales, addressing limitations of overlapping structures and improving performance in heterogeneous crime environments.

**Neural-Based Crime Prediction** Neural approaches like GNNs Chai et al. (2022), ST-GCNs Yu et al. (2017), and Transformers Vaswani et al. (2017) have advanced crime forecasting but face limitations: they require predefined spatial structures, struggle with data skew across density variations, and suffer from quadratic computational complexity. SMART-CARE overcomes these through adaptive partitioning that dynamically adjusts to local variance, enabling efficient hierarchical refinement with linear scalability while maintaining accuracy across diverse urban regions.

Despite advancements in crime prediction, existing methods struggle with imbalanced, high-density datasets, hierarchical spatial relationships, and scalability. Techniques like R-trees and clustering often fail to deliver fine-grained predictions in complex crime environments. To address these, we propose SMART-CARE, a scalable framework integrating dynamic spatial segmentation, hierarchical modelling, and efficient learning mechanisms. SMART-CARE leverages adaptive partitioning and parent-child model refinement to balance data distributions and capture spatio-temporal dynamics, enhancing prediction accuracy. This section details the design and implementation of SMART-CARE, outlining its components and operational workflow.

## 3 METHODOLOGY

We introduce the **SMART-CARE** framework, comprising **S**patio-**M**edian **A**daptive **R**ecursive **T**ree (SMART-QT) for adaptive spatial partitioning and **C**rime **A**daptive **R**efined **E**nsemble (CARE) for hierarchical prediction refinement. SMART-QT partitions urban spaces using variance-aware median splitting, while CARE enables fine-grained forecasting through feature propagation and model inheritance across spatial scales.

Given a dataset $\mathcal{D} = \{(\mathbf{x}_i, t_i, c_i)\}_{i=1}^{N}$ where $\mathbf{x}_i \in \mathbb{R}^d$ contains spatio-temporal features derived from timestamp $t_i$, and $c_i \in \mathbb{N}$ is the daily crime count, our objective is to learn a predictor $f : \mathbf{x}_i \mapsto \hat{c}_i$. The features $\mathbf{x}_i$ include spatial coordinates (longitude, latitude) and temporal encodings (cyclical hour/month, lagged counts, rolling statistics) extracted from $t_i$. SMART-QT recursively partitions the spatial domain $\mathcal{S} \subset \mathbb{R}^2$ into nodes $\{\mathcal{N}_j\}$ using median splits constrained by adaptive capacity $T_{\max}$ and depth limit $L_{\max}$. Each node $\mathcal{N}_j$ contains a subset $\mathcal{D}_j \subset \mathcal{D}_{\text{train}}$ and trains a local regressor $f_j$ refined from its parent model. Test samples are routed to appropriate leaves for prediction using the corresponding $f_j$.

## 3.1 SMART QUADTREE

The classical quadtree recursively partitions a 2D space into four quadrants when a node contains more than a fixed threshold $M$ points, with recursion bounded by maximum depth $L$ (Appendix A.1: Traditional Quadtree). Formally, subdivision occurs if $n > M$, where $n$ is the point count in a node. However, this static approach fails to adapt to heterogeneous urban crime data with dense hotspots and sparse regions. We propose the **SMART Quadtree** (Spatio-Median Adaptive Recursive Tree) with four key innovations: (1) adaptive node capacity thresholds derived from data statistics, (2) adaptive depth limits scaling with dataset size, (3) median-based spatial splitting for balanced partitions, and (4) merging of sparse leaf nodes to prevent over-fragmentation (Algorithm 2:SMART Quadtree Construction).

**Adaptive subdivision threshold ($T_{\max}$):** SMART-QT replaces a fixed node capacity with a variance-aware, dataset-scaled threshold:

$$T_{\max} = \left\lceil \max\left(T_{\min}, \min\left(\frac{\beta}{1 + \sigma^2/\gamma} + \frac{|P|}{\delta}, T_{\text{cap}}\right)\right)\right\rceil, \tag{1}$$

where $\sigma^2$ is the (sample) variance of crime counts in node $P$ and $|P|$ its point count. The lower/upper clamps $T_{\min}$ and $T_{\text{cap}}$ are computed by scaling the reference parameters $(\alpha, \kappa)$ to the dataset size (detailed in: Dynamic Parameter Scaling). Intuitively, the $\beta/(1 + \sigma^2/\gamma)$ term reduces $T_{\max}$ in high-variance (hotspot) nodes, while the $|P|/\delta$ term raises the threshold in already dense nodes to avoid excessive splitting. Clipping with $T_{\min}$ and $T_{\text{cap}}$ prevents pathological values in extreme regions. All parameters $(\alpha, \beta, \gamma, \delta, \kappa, \lambda)$ are auto-scaled by dataset size (Appendix A.2).

**Adaptive Tree Depth ($L_{\max}$):** Unlike classical quadtrees that enforce a globally fixed maximum depth, SMART-QT dynamically determines the allowable depth for each subtree based on the dataset size and localised variance in crime patterns. Specifically, the maximum depth is computed as

$$L_{\max} = \min\left(\eta \cdot \log_2(n_{\text{total}}), \ \log_2(n_{\text{total}}) + 1 + \sigma_{\text{local}}^2\right), \tag{2}$$

where $n_{\text{total}}$ is the total number of points in the dataset and $\sigma_{\text{local}}^2$ denotes the sample variance of crime counts within the node (computed with $ddof = 1$). The scaling factor $\eta$ (e.g., $\eta = 1.5$) modulates global complexity. The result is then rounded up to an integer and clamped to a practical range $L_{\min} \leq L_{\max} \leq L_{\text{cap}}$ (e.g., $L_{\min} = 5$, $L_{\text{cap}} = 15$) to prevent overfitting or under-representation. This adaptive rule yields deeper trees in high-variance regions while constraining depth in homogeneous areas, balancing expressiveness and efficiency by tailoring quadtree granularity to both global data volume and local heterogeneity.

**Median-Based Splitting and Point Retention:** To ensure balanced spatial partitions, SMART-QT computes the median of the longitude and latitude values from all points $P$ in the node. The median coordinates $\mathbf{x}_{\text{mid}} = (x_{\text{mid}}[0], x_{\text{mid}}[1])$ are: $x_{\text{mid}}[0] = \text{median}(\mathbf{x}_i[0] \mid \mathbf{x}_i \in P)$, $x_{\text{mid}}[1] = \text{median}(\mathbf{x}_i[1] \mid \mathbf{x}_i \in P)$. The split produces four axis-aligned quadrants $(N_{\text{NW}}, N_{\text{NE}}, N_{\text{SW}}, N_{\text{SE}})$. Unlike classical quadtrees that remove points from the parent, SMART-QT *retains* all parent points to enable parent–child knowledge transfer. Each child node is assigned a reference (index set) to the subset of parent points falling in its quadrant, rather than duplicating the underlying data. This reference-based redistribution preserves upper-level context for hierarchical modelling while avoiding unnecessary memory copies (see section 3.2 for implementation details).

**Strategic Small-Leaf Merging:** Classical quadtrees often over-partition sparse or low-variance regions, creating small, noisy leaves that reduce generalisation and increase computational cost. To

mitigate this, SMART-QT integrates a variance-aware *small-leaf merging* mechanism. A node $\mathcal{N}$ with point set $P$ and area $A$ (density $\rho = |P|/A$) is marked for merging if (i) $0 < |P| < \tau$ (minimum threshold), or (ii) $\rho$ is a density outlier by the IQR rule ($\rho < Q_1 - \phi \cdot \text{IQR}$ or $\rho > Q_3 + \phi \cdot \text{IQR}$, $\phi = 1.5$). Each candidate $\mathcal{N}_c$ is paired with a sibling $\mathcal{N}_s$ (same parent) that satisfies $|P_c| + |P_s| < \tau_{\max}$ ($\tau_{\max} = 2.5\tau$) and minimises $|\rho_c - \rho_s|$. After merging, all points are reassigned to $\mathcal{N}_s$, $\mathcal{N}_c$ is deactivated, and merged pairs are tracked. This reduces fragmentation while preserving coherent, reliable partitions. (Full merging pseudocode and geometric examples are in Appendix A.3.)

**Periodic Threshold Re-tuning:** To ensure that the quadtree remains adaptive across varying spatial resolutions, SMART-QT periodically re-tunes the node capacity $T_{\max}$ and maximum depth $L_{\max}$ at designated levels, reflecting localised variations in crime density. This re-tuning is triggered at nodes where the depth satisfies `node_level` mod $\nu = 0$, with $\nu = 1$ for large datasets (e.g., $n_{\text{total}} > 1,000,000$), enabling threshold recalibration at alternate levels. The hyperparameter $\nu \in \mathbb{Z}^+$ controls the frequency of threshold re-tuning and defines the re-tuning interval across tree depth levels. Refer example in Appendix A.4.

**Spatio-Temporal Feature Propagation:** Unlike classical quadtrees that store only spatial coordinates $(x_i, y_i)$, SMART-QT attaches a rich spatio-temporal feature vector $\mathbf{x}_i \in \mathbb{R}^d$ to each point so that every node can learn with full contextual information. Features include spatial coordinates, temporal attributes derived from the raw timestamp $t_i$ (hour, day-of-week, month), cyclic encodings (e.g. $\text{hour}_{\sin} = \sin(2\pi \cdot \text{hour}/24)$), lagged crime counts $c_{i-k}$ for $k \in \{1, 2, 3\}$, and short rolling statistics (e.g. 7-day mean). All features are standardised (zero mean, unit variance) and retained as points are referenced by parent and child nodes; in addition, we propagate the parent prediction via a dedicated `Prediction` column. By combining inherited context (parent prediction and features) with local data, each node specialises its model to capture recurring hotspots, seasonal patterns, and local variance. (Further implementation details and feature lists are in Appendix A.5.)

**Dynamic Parameter Scaling:**

To generalise SMART-QT across datasets of varying sizes, all key hyperparameters are dynamically scaled based on the total number of points in the dataset, denoted by $n_{\text{total}}$. Using a reference dataset size $n_{\text{ref}}$, a scaling factor is computed as $s = \frac{n_{\text{total}}}{n_{\text{ref}}}$, which proportionally adjusts parameters such as $\alpha$, $\beta$, $\kappa$, and `min_base`. Here, `min_base` represents a lower-bound cap for the node capacity in the adaptive threshold function (Eq. 1), ensuring that even in low-variance or low-density regions, each node retains a minimum number of points before being split. This prevents over-fragmentation in sparse areas. These control node capacity and subdivision thresholds. For parameters that benefit from smoother adaptation, such as $\lambda, \gamma$, and $\delta$, a logarithmic scaling is applied to avoid abrupt changes as the dataset size grows. The scaled values are computed as:

$$\lambda = \text{clip}(\lambda_0 \cdot \log(1+s), 5, 20), \quad \gamma = \text{clip}(\gamma_0 \cdot \log(1+s), 1, 5), \quad \delta = \text{clip}(\delta_0 \cdot \log(1+s), 1, 5). \quad (3)$$

where $\text{clip}(\cdot)$ ensures values remain within practical bounds. This dynamic scaling ensures SMART-QT remains robust and consistently adaptive without requiring manual re-tuning when deployed across datasets of vastly different sizes. The complete recursive construction process of SMART-QT is outlined in Algorithm 2. This adaptive and recursive construction makes SMART-QT particularly suitable for heterogeneous crime landscapes, as it dynamically balances spatial resolution with statistical complexity, leading to more granular modelling in high-variance areas while avoiding over-fitting in low-density regions (implementation details and tuned defaults are given in Appendix A.6).

### 3.2 CARE PREDICTIVE MODEL

CARE (**C**rime **A**daptive **R**efined **E**nsemble) is a hierarchical prediction scheme that fine-tunes local regressors at each quadtree node using (i) feature-propagation — the parent's prediction appended as a feature to child inputs, and (ii) model-inheritance — warm-starting the child regressor from its parent's model and fine-tuning on local data. These two mechanisms jointly enable fast local adaptation and improved accuracy in sparse and dense regions. (Appendix B.1: CARE-PM Algorithm).

#### 3.2.1 HIERARCHICAL MODEL TRAINING

Figure 1 illustrates the SMART-CARE framework. After constructing the SMART-QT quadtree, CARE-PM trains node models in breadth-first (top-down) order so that each parent is fitted before its children. Let node $\nu$ have training set $\mathcal{D}_\nu = \{(\mathbf{x}_i, c_i)\}_{i \in I_\nu}$ and parent $\pi(\nu)$. The local regressor

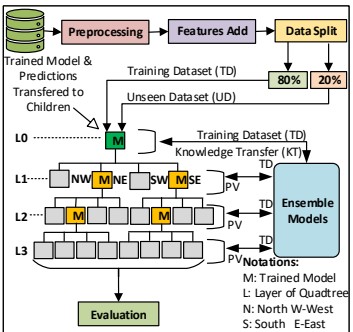

Figure 1: SMART-CARE Framework.

is $f_\nu(\mathbf{x}; \theta_\nu)$. Using feature propagation, the augmented input is $\mathbf{x}_i^{\text{aug}} = [\mathbf{x}_i, \ f_{\pi(\nu)}(\mathbf{x}_i)]$, (where $f_{\pi(\nu)}$ is absent at the root). Each node solves a regularized local objective:

$$\theta_\nu^\star = \arg\min_\theta \frac{1}{|\mathcal{D}_\nu|} \sum_{(\mathbf{x},c) \in \mathcal{D}_\nu} \ell\big(f_\nu(\mathbf{x}^{\text{aug}}; \theta), c\big) + \lambda_\nu \, \Omega(\theta, \theta_{\pi(\nu)}), \tag{4}$$

where $\ell(\cdot, \cdot)$ is the sample loss (MSE) and $\Omega$ is a proximity regularizer encouraging the child to remain close to the parent model when appropriate. For the neural variant we use $\Omega(\theta, \theta_\pi) = \|\theta - \theta_\pi\|_2^2$. For the tree (XGBoost) variant, proximity is implemented via warm-start (pass the parent booster and optionally restrict the number of new trees $T_{\text{new}}$ or lower the learning rate).

**Root Node Training:** The root node, encompassing the entire spatial domain, has no parent, so CARE begins by establishing an initial predictor to capture global crime trends. A baseline model $f_{\text{base}}$ (either a lightweight regressor or a shallow NN) is trained on the entire training set $\mathcal{D}_{\text{train}}$ using the spatio-temporal feature vectors $\mathbf{x}_i \in \mathbb{R}^d$, minimizing the mean squared error $\min_\theta \sum_{i \in \mathcal{D}_{\text{train}}} \big(y_i - f_{\text{base}}(\mathbf{x}_i; \theta)\big)^2$. This baseline produces preliminary global crime-count estimates $\hat{y}_{\text{base}}(i) = f_{\text{base}}(\mathbf{x}_i)$. To stabilise learning across different feature scales, these estimates are standardized to zero mean and unit variance as $\hat{y}'_{\text{base}}(i) = \dfrac{\hat{y}_{\text{base}}(i) - \mu_{\hat{y}_{\text{base}}}}{\sigma_{\hat{y}_{\text{base}}}}$, with $\mu_{\hat{y}_{\text{base}}}$ and $\sigma_{\hat{y}_{\text{base}}}$ computed over $\mathcal{D}_{\text{train}}$. The standardized predictions $\hat{y}'_{\text{base}}(i)$ are then appended as an auxiliary feature ("Prediction") to form augmented inputs $\mathbf{x}_{i,\text{aug}} = [\mathbf{x}_i, \hat{y}'_{\text{base}}(i)]$. *Tree-based variant (XGBoost):* The root's final regressor $f_{\text{root}}$ is an XGBoost booster trained on $\{(\mathbf{x}_{i,\text{aug}}, y_i)\}$, minimising squared error. *NN variant:* The root model $f_{\text{root}}$ is a feedforward NN initialised randomly and trained on the same augmented inputs, with an optional regulariser to improve generalisation. In both cases, the root's refined predictions $\hat{y}_1(i) = f_{\text{root}}(\mathbf{x}_{i,\text{aug}})$ serve as the top-level forecasts, forming the foundation for child-node refinement. (Appendix: B.2 for more details).

**Child Node Training:** For each child node $\nu$, CARE trains a local regressor $f_\nu$ on its dataset $\mathcal{D}_\nu \subset \mathcal{D}_{\text{train}}$, defined by the SMART-QT partition. The input features include the parents' predictions $\hat{y}_{\pi(\nu)}(i)$, standardised within $\mathcal{D}_\nu$ to have zero mean and unit variance, which are appended as an additional feature: $\mathbf{x}_{i,\text{aug},\nu} = [\mathbf{x}_i, \hat{y}'_{\pi(\nu)}(i)]$. The child model is then trained to minimise the local mean squared error: $\hat{y}_\nu(i) = f_\nu(\mathbf{x}_{i,\text{aug},\nu})$, $\qquad \theta_\nu^\star = \arg\min_\theta \frac{1}{|\mathcal{D}_\nu|} \sum_{i \in \mathcal{D}_\nu} (y_i - \hat{y}_\nu(i))^2$. To accelerate convergence and ensure consistency, $f_\nu$ is warm-started from the parent model $f_{\pi(\nu)}$. This recursive refinement continues down the hierarchy, with each node specialising its predictions to local spatial regions while inheriting global context from its ancestors. (Appendix: B.3)

**Prediction Refinement Across Levels:** At each node $\nu$ in the SMART-QT hierarchy, the CARE Predictive Model refines predictions using the parent's inherited estimate $\hat{y}_{\pi(\nu)}(i) = f_{\pi(\nu)}(\mathbf{x}_{i,\text{aug}})$ as a contextual input feature. After training the node's XGBoost regressor $f_\nu$ on $\mathcal{D}_\nu \subset \mathcal{D}_{\text{train}}$, the stored predictions are updated to $\hat{y}_\nu(i) = f_\nu(\mathbf{x}_{i,\text{aug},\nu})$, replacing the parent's estimate, and these refined predictions are passed to descendant nodes. This recursive process, initialized by the root's baseline $\hat{y}_{\text{base}}(i) = f_{\text{base}}(\mathbf{x}_i)$, progresses through levels, with each $f_\nu$ learning the residual errors of its parent by minimizing $\sum_{i \in \mathcal{D}_\nu} (y_i - \hat{y}_\nu(i))^2$. The refinement can be expressed as a hierarchical ensemble sequence $\hat{y}_0 \to \hat{y}_1 \to \cdots \to \hat{y}_{\leq L_{\max}}$, where $\hat{y}_k(i)$ improves upon $\hat{y}_{k-1}(i)$ and the process terminates in the leaf-level prediction $\hat{y}_{\text{final}}(i)$. If a node remains unsplit, its $\hat{y}_\nu(i)$ is final;

otherwise, descendants further refine it. This step-wise approach integrates global patterns from higher levels with local deviations at lower levels, enhancing prediction accuracy across diverse crime distributions.

### 3.3 Optimised Training and Inference:

**Efficient training and inference:** CARE-PM accelerates training via *parent→child model inheritance*: each child model $f_\nu$ is warm-started from its parent $f_{\pi(\nu)}$ (e.g., parent XGBoost booster or NN weights), enabling faster convergence and higher accuracy. Hyperparameter tuning is performed only at the root, avoiding expensive grid searches at every node. At inference, a sample activates at most one model per tree level. Let $L_{\max}$ denote the quadtree depth, $p$ the feature dimension, and $k$ the number of trees in an XGBoost booster. The per-node inference cost is $O(p \cdot k)$ for XG-Boost, and $O(\sum_{\ell=1}^{H} d_{\ell-1} d_\ell)$ for an $H$-layer neural network with layer widths $\{d_\ell\}$ (dominated by matrix multiplications). In both cases, once the architecture or booster is fixed after tuning, the per-node cost is effectively constant. Thus, the per-sample inference cost scales as $O(L_{\max})$, and since quadtree depth grows logarithmically with dataset size $n$, the overall complexity is $O(\log n)$ under the fixed-model assumption. Empirical timings in Appendix B.5 confirm that the constant factor is small in practice.

**Handling Sparse or Merged Nodes:** CARE-PM addresses sparse nodes in the SMART-QT hierarchy by skipping or merging regions with insufficient data. If a leaf node $\nu$ has $|\mathcal{D}_\nu| < \tau$ and no sibling to merge with, CARE skips training, using the parent's predictions $\hat{y}_{\pi(\nu)}(i) = f_{\pi(\nu)}(\mathbf{x}_{i,\text{aug}})$ as the final output. Alternatively, small sibling leaf nodes are merged into a single unit by combining their data, ensuring $|\mathcal{D}_{\text{combined}}| \leq \kappa$, then training on the regressor model. Merged nodes are marked, with one sibling's model covering both areas, trained like other nodes. This ensures sufficient sample sizes, reducing variance and stabilising predictions in low-density regions.

**Hierarchical Prediction:** SMART-CARE performs inference through hierarchical refinement, where predictions are sequentially enhanced by models at decreasing spatial scales. This top-down process ensures forecasts incorporate both global trends and local patterns, detailed in Appendix B.6.

## 4 Experiments and Results

**Experimental Setup:** We evaluate SMART-QT and CARE-PM on NYC (7.8M records, 2008–2023) and Chicago (8.2M records, 2001–2024) crime datasets, using spatio-temporal features (Date-Time, Latitude, Longitude). Models are trained in a breadth-first traversal, where parent predictions are propagated to children for hierarchical refinement (Appendix D.1 for details).

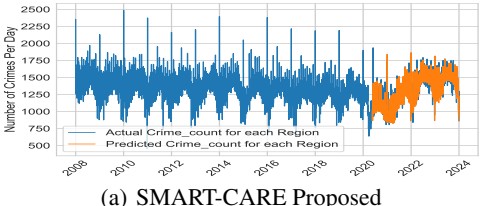
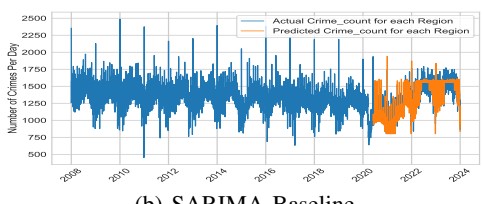

(a) SMART-CARE Proposed        (b) SARIMA-Baseline

Figure 2: NYC Time Series Prediction Analysis

**Evaluation Metrics:** We report Mean Absolute Error (MAE), Root Mean Squared Error (RMSE), and Adjusted $R^2$ as primary metrics. MAE captures average errors, RMSE emphasizes larger deviations, and Adjusted $R^2$ accounts for predictor count, jointly reflecting predictive accuracy and generalization across spatio-temporal scales.

**Time Series:** Figure 2 (a) demonstrates that CARE-PM effectively captures daily crime trends, including upper and lower bounds and seasonality, outperforming SARIMA Butt et al. (2021), which struggles with these patterns. By leveraging spatio-temporal features and hierarchical refinement, CARE-PM better adapts to temporal fluctuations across regions, enhancing prediction reliability over traditional time-series models, especially in datasets with complex crime dynamics like NYPD and Chicago.

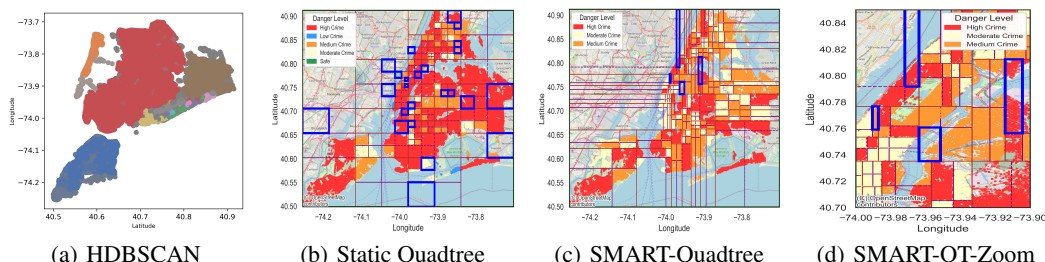

(a) HDBSCAN     (b) Static Quadtree     (c) SMART-Quadtree     (d) SMART-QT-Zoom

Figure 3: NYC Crime Heatmap: HDBSCAN, Static & Adaptive-QT

**Heatmap Comparison:** Figure 3 compares partitioning strategies: HDBSCAN produces irregular clusters, standard Quadtree shows sparse nodes (blue rectangles), while SMART-QT achieves balanced distribution. Our method reduces nodes by 35% in sparse regions while maintaining hotspot granularity, validating adaptive partitioning superiority (Appendix: D.2.

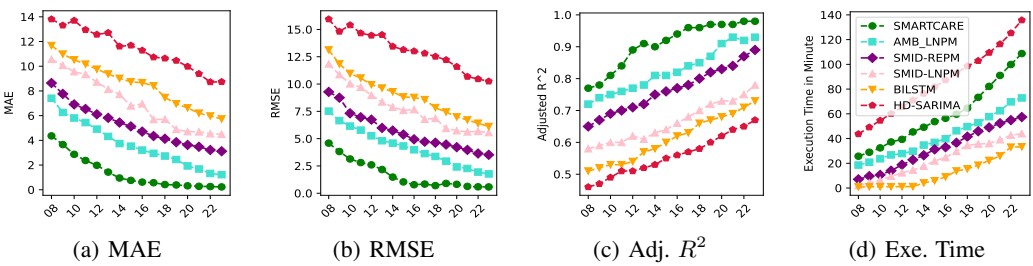

(a) MAE     (b) RMSE     (c) Adj. $R^2$     (d) Exe. Time

Figure 4: NYC Temporal Comparative Analysis (Tree-Based Model).

**Ablation Variant Comparison:** An ablation study (Appendix: C) confirms SMART-CARE's advantages over three variants: SMID-LNPM (MAE=5.26) and AMB-LNPM (MAE=2.36), which lack hierarchical knowledge transfer, and SMID-REPM (MAE=3.23), which uses static midpoint splitting. SMART-CARE achieves superior performance (MAE=0.92) by combining adaptive median-based partitioning with parent-to-child model refinement, effectively capturing complex spatio-temporal patterns (Figure 4) (For more detail Appendex: D.3).

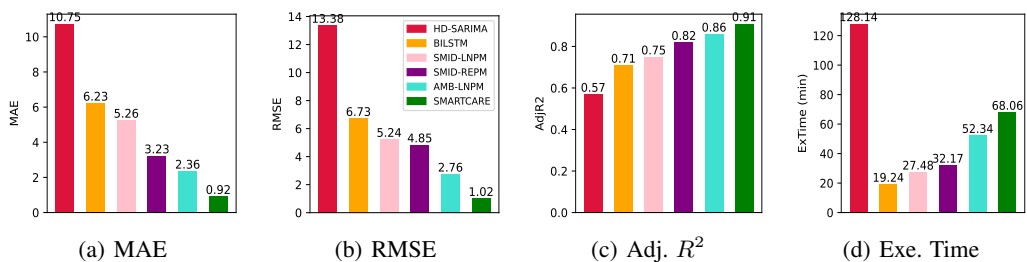

(a) MAE     (b) RMSE     (c) Adj. $R^2$     (d) Exe. Time

Figure 5: Avg. Comparative Analysis of Various Frameworks (Tree-Based Model).

**Temporal Comparative Analysis:** SMART-CARE was evaluated on incrementally aggregated yearly crime data (2008-2023) using $D_t = \bigcup_{i=1}^{t} Y_i$. As the dataset size increased, error rates decreased, demonstrating improved pattern recognition. SMART-CARE demonstrates flexibility by performing effectively with both tree-based and neural model instantiations. The tree-based implementation achieves MAE=0.23 and Adjusted R²=0.94 on aggregated data (2008-2023), significantly outperforming literature baselines Butt et al. (2021; 2024) (MAE: 8.74, 6.12). Neural

implementations (Fig. 6) also show competitive performance across GRU, LSTM, BiLSTM, and MLP architectures while maintaining computational efficiency (Figure 4).

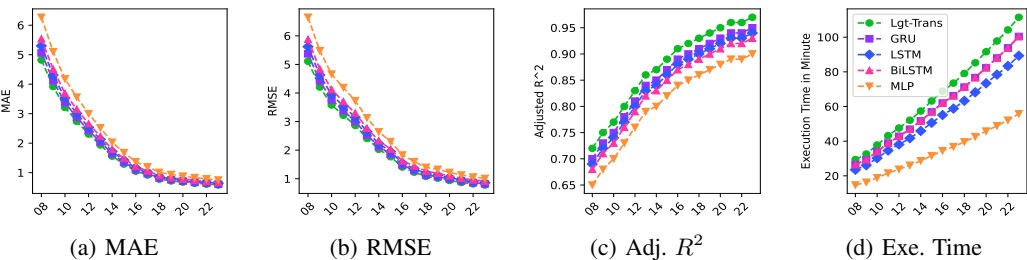

| (a) MAE | (b) RMSE | (c) Adj. $R^2$ | (d) Exe. Time |

Figure 6: Chicago Temporal Comparative Analysis (NN-Based Model).

Figure 5 shows the average MAE across each year for various frameworks, based on the results presented in Figure 4. SMART-CARE significantly outperforms the baseline model Butt et al. (2021) and ablation variants, reducing the average MAE from 10.75 to 0.92. By combining spatiotemporal analysis, hierarchical knowledge transfer from parent node models, and predictive learning through the quadtree structure, SMART-CARE effectively captures complex crime patterns, leading to substantial improvement in prediction accuracy.

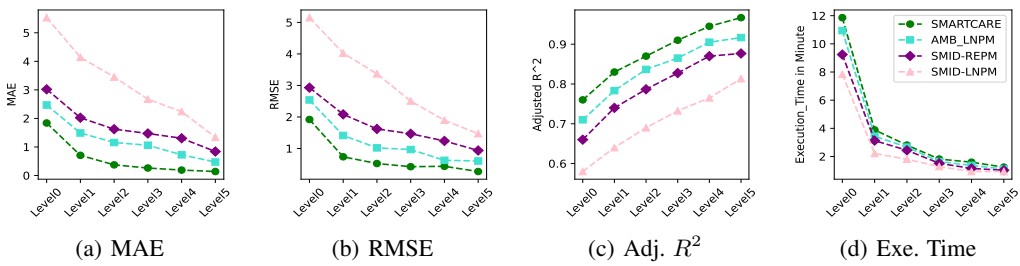

| (a) MAE | (b) RMSE | (c) Adj. $R^2$ | (d) Exe. Time |

Figure 7: Spatial Comparative Analysis of Various Frameworks (Tree-Based Model).

**Spatial Comparative Analysis:** To evaluate performance across hierarchical geographical scales, we compared SMART-CARE, AMB-REPM, SMID-REPM, and SMID-LNPM on a 10-year dataset. For SMID-REPM and SMID-LNPM, we fixed node capacity at $T_{\max} = 500,000$ and maximum depth at $L_{\max} = 5$, while SMART-CARE and AMB-REPM used adaptive thresholds and depth, per SMART-QT's design. Error metrics (MAE, RMSE, Adjusted $R^2$) were aggregated per quadtree node, averaging errors at each layer (e.g., layer-0, layer-5). Figure 7 shows SMART-CARE outperformed others, with avg. MAE decreasing from 1.64 (layer-0, root node with one model) to 0.13 (layer-5, averaging over multiple granular models), compared to 6.46 to 0.56 for other models, highlighting its adaptive structure's ability to capture fine-grained spatial features as depth increases.

Table 1: Performance Metrics Comparison

| Metric | SMART-CARE | AMB_LNPM | SMID-REPM | SMID-LNPM |
|---|---|---|---|---|
| Range Query Time (s) | 0.0125 | 0.0140 | 0.0165 | 0.0172 |
| Points Found in Query | 1548 | 1548 | 1420 | 1385 |
| Memory Usage (MB) | 38.61 | 40.20 | 71.75 | 63.00 |

Table 2: Density Distribution Comparison

| Metric | SMART-CARE | AMB_LNPM | SMID-REPM | SMID-LNPM |
|---|---|---|---|---|
| Count | 66 | 66 | 112 | 112 |
| Mean | 6169219.41 | 6200000.00 | 7020000.00 | 7200000.00 |
| Std. Dev. | 5982675.80 | 6100000.00 | 7400000.00 | 7600000.00 |
| Min | 142360.71 | 145000.00 | 180000.00 | 170000.00 |
| 25th Percentile | 1656491.97 | 1700000.00 | 1950000.00 | 2000000.00 |
| Median | 3128489.60 | 3150000.00 | 3750000.00 | 3900000.00 |
| 75th Percentile | 9659166.74 | 9700000.00 | 9800000.00 | 9900000.00 |
| Max | 21207441.01 | 21500000.00 | 23500000.00 | 24000000.00 |

**Feature Importance Analysis:** Table 3 highlights the importance scores of spatio-temporal and predictive features across NYC (7.9M rows) and Chicago (8.2M rows) datasets. For NYC, `Prediction` (0.538) is the most significant, reflecting autocorrelation from hierarchical knowledge transfer in SMART-CARE, followed by `Month` (0.150) for seasonality and `Lag1` (0.116) for short-term trends. For Chicago, `Prediction` (0.502) leads, with `Month` (0.245)

Table 3: Feature Importance Score.

| | NYC 7.9 M. Rows | | CHICAGO 8.2 M. Rows |
|---|---|---|---|
| Feature | Importance | Feature | Importance |
| Prediction | 0.538 | Prediction | 0.502 |
| Month | 0.150 | Month | 0.245 |
| Lag1 | 0.116 | Lag1 | 0.212 |
| Lag2 | 0.115 | Lag2 | 0.033 |
| RollMean7d | 0.078 | RollMean7d | 0.006 |
| Day | 0.003 | Day | 0.002 |

Table 4: QT Structural Metrics Comparison

| Metric | SMART-CARE | AMB_LNPM | SMID-REPM | SMID-LNPM |
|---|---|---|---|---|
| Max Depth | 3 | 3 | 2 | 2 |
| Total Nodes | 66 | 66 | 112 | 112 |
| Leaf Nodes | 49 | 49 | 86 | 86 |
| Avg. Points per Leaf | 8259.86 | 8657.52 | 13524.84 | 12458.73 |
| Variance (Leaf) | 5523618.82 | 5632654.00 | 6135974.00 | 6354297.00 |
| Merged Nodes | 4 | 4 | 26 | 26 |
| Empty Nodes | 0 | 0 | 14 | 16 |
| Points in Merged Nodes | 32830 | 34521 | 98211 | 10542 |

and `Lag1` (0.212) showing stronger temporal influence. Spatial features (`Scl_Longitude`, `Scl_Latitude`) and `Date` have minimal impact (not listed, <0.05), as SMART-QT handles location. `Lag2` and `RollMean7d` contribute modestly (e.g., 0.115, 0.078 for NYC), aiding sparse region predictions. This validates SMART-CARE's temporal focus.

Table 5: SMART-CARE Hyp-Tune.

| Parameter | Value | MAE | RMSE | Adj. $R^2$ |
|---|---|---|---|---|
| | 2000 | 0.25 | 0.35 | 0.95 |
| $\alpha$ | 5000 | 0.40 | 0.55 | 0.90 |
| | 10000 | 0.43 | 0.58 | 0.89 |
| | 50000 | 0.26 | 0.36 | 0.94 |
| $\kappa$ | 100000 | 0.38 | 0.52 | 0.91 |
| | 150000 | 0.42 | 0.56 | 0.89 |
| | 50000 | 0.24 | 0.34 | 0.96 |
| $\beta$ | 10000 | 0.42 | 0.57 | 0.89 |
| | 25000 | 0.39 | 0.53 | 0.91 |
| | 2 | 0.27 | 0.37 | 0.93 |
| $\delta$ | 5 | 0.37 | 0.50 | 0.92 |
| | 10 | 0.41 | 0.54 | 0.90 |
| | 5000 | 0.25 | 0.35 | 0.95 |
| min_base | 2000 | 0.39 | 0.54 | 0.90 |
| | 10000 | 0.41 | 0.57 | 0.88 |

Table 6: Nueral Fine-tuning Hyperparameters

| Model | Optimizer | Root LR | Child LR | Dropout | Batch Size | Epochs |
|---|---|---|---|---|---|---|
| Light Transformer | AdamW | $1 \times 10^{-4}$ | $5 \times 10^{-5}$ | 0.2–0.3 | 64 | 15–20 |
| GRU | Adam | $5 \times 10^{-3}$ | $1 \times 10^{-3}$ | 0.1–0.2 | 128 | 10–15 |
| LSTM | Adam | $3 \times 10^{-3}$ | $8 \times 10^{-4}$ | 0.2 | 128 | 10–15 |
| BiLSTM | Adam | $3 \times 10^{-3}$ | $8 \times 10^{-4}$ | 0.3 | 128 | 10–15 |
| MLP | Adam | $1 \times 10^{-3}$ | $5 \times 10^{-4}$ | 0.1 | 256 | 8–12 |

Table 7: Neural Learning-rate Schedules

| Model | Scheduler | Warm-up | Notes |
|---|---|---|---|
| Light Transformer | Cosine Annealing | 2 epochs | LR decays to $1 \times 10^{-6}$ |
| GRU | StepLR / ReduceLROnPlateau | 1 epoch | $\gamma = 0.5$ every 5 epochs |
| LSTM | StepLR / ReduceLROnPlateau | 1 epoch | $\gamma = 0.5$, weight decay 1e-4 |
| BiLSTM | StepLR | 1 epoch | Stronger regularization, $\gamma = 0.5$ |
| MLP | ReduceLROnPlateau | None | Monitor val loss, factor 0.5 |

**Hyper-Parameter Comparative Analysis:** We evaluated SMART-CARE's hyper-parameters ($\alpha$, $\kappa$, $\beta$, $\delta$, min_base) for adaptive quadtree balancing. Table 5 shows tuned values (e.g., $\alpha = 2000$, $\beta = 50000$) achieving optimal MAE (0.24–0.27), RMSE (0.34–0.37), and Adjusted $R^2$ (0.93–0.96), compared to alternatives (e.g., $\alpha = 10000$, MAE: 0.43). A table, confirming that tuned parameters enhance prediction accuracy and quadtree balance across diverse datasets.

The hyperparameters and learning-rate schedules in Tables 6 and 7 support hierarchical quadtree fine-tuning for crime prediction. Higher root learning rates (e.g., $1 \times 10^{-4}$ for Light Transformer, $5 \times 10^{-3}$ for GRU) enable global learning, while lower child rates (e.g., $5 \times 10^{-5}$, $1 \times 10^{-3}$) stabilize local refinements. Dropout scales with complexity (0.2–0.3 for Transformers/BiLSTM, 0.1 for GRU/MLP). Batch sizes and epochs adjust for model size and data (e.g., 8–12 for MLP, 10–15 for LSTM). Schedules include a 2-epoch warm-up with cosine annealing for Transformers (to $1 \times 10^{-6}$), 1-epoch warm-up with step decay ($\gamma = 0.5$) for RNNs, and validation-based decay (factor 0.5) for MLPs, ensuring efficient transfer, convergence, and stability.

**Quadtree Analysis Summary:** We evaluated SMART-CARE's quadtree efficiency through structural, performance, and density metrics. Table 4 shows SMART-CARE with a max depth of 3, 66 nodes (49 leaf nodes), and lower variance (5523618.82) than AMB-LNPM, SMID-REPM, and SMID-LNPM, with 4 merged nodes reducing points to 32830. Table 1 highlights a 0.0125s range query time and 38.61 MB memory usage, outperforming others. Table 2 indicates a balanced density (mean: 6169219.41, median: 3128489.60), confirming SMART-CARE's adaptability.

## 5 CONCLUSION

We presented SMART-CARE, an adaptive quadtree framework that dynamically partitions urban spaces using variance-aware median splitting and hierarchical model refinement. Our approach addresses key challenges in crime prediction: handling data skew through strategic small-leaf merging, capturing multi-scale patterns via feature propagation and model inheritance, and maintaining computational efficiency through periodic re-tuning. Extensive evaluation on NYC and Chicago crime data demonstrates SMART-CARE's superiority over uniform grids, static quadtrees, and recent baselines, achieving MAE reductions to 0.92 (average) and 0.23 (best-case) while reducing execution time by 53%. The framework's flexibility across both tree-based and neural instantiations highlights its generalizability for spatio-temporal forecasting tasks. Future work will explore real-time adaptation and multi-modal data integration.

**Reproducibility Statement:** We have taken several measures to ensure the reproducibility of our work. The datasets used (NYPD Complaint Data and Chicago Crime Data) are publicly available and cited in Section A.6. Data preprocessing steps, including spatio-temporal feature extraction and quadtree construction, are detailed in Section A and Appendix A.5. The complete recursive algorithm for SMART-QT, dynamic parameter scaling, and periodic re-tuning is presented in Appendix 2, along with sensitivity analyses of hyperparameters. CARE-PM training and inference procedures are formalized in Section B and Appendix B.6, with additional notes on handling sparse nodes and prediction refinement. Hyperparameter settings for XGBoost and neural variants are reported in Table B.4, and empirical timing results are provided in Appendix B.5. Finally, we provide an anonymous repository with full source code, preprocessing scripts, and trained models to facilitate exact reproduction of all results.

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

# A  SMART QUADTREE

## A.1  TRADITIONAL QUADTREE

The quadtree begins with the entire spatial area, $A$, and recursively subdivides it into four quadrants if the number of the data points ($N_A$), within a node exceeds the predefined node capacity ($\mathcal{N}_{\text{capacity}}$), the region $A$ is divided into four equally sized sub-regions: Northwest (NW), Northeast (NE), Southwest (SW), and Southeast (SE). This process continues until all nodes satisfy $N_A \leq \mathcal{N}_{\text{capacity}}$, resulting in a hierarchical segmentation of the spatial environment.

To better understand the functionality of our quadtree structure, refer to Algorithm (**??**) part-I and Figure 8, which represent a quadtree containing fifteen data points (blue dots), each signifying a crime incident within its spatial boundary. In this example, the node capacity is set to $\mathcal{N}_{\text{capacity}} = 2$, and the maximum quadtree depth is limited to $\mathcal{L}_{\text{capacity}} = 10$.

**Step 1: Quadtree Initialisation :** The quadtree begins with the root node ($\mathcal{N}_{\text{root}}$), encompassing the entire study area, as shown in Figure 8 (green dotted line). The root node's boundary is defined as $R(X_{\min}, Y_{\min}, X_{\max}, Y_{\max})$, where $X_{\min}, Y_{\min}, X_{\max}, Y_{\max}$ represent the dataset's spatial extent. The quadtree is initialized as: $Q(R(X_{\min}, Y_{\min}, X_{\max}, Y_{\max}), \mathcal{N}_{\text{capacity}}, \mathcal{L}_{\text{capacity}})$, where $\mathcal{N}_{\text{capacity}}$ specifies the maximum points per node, and $\mathcal{L}_{\text{capacity}}$ limits the tree depth to ensure computational

---

**Algorithm 1** Traditional Quadtree Algorithm

---

**Require:** Point object $p$ to be inserted, boundary rectangle $R$, node capacity $\mathcal{N}_{\text{capacity}}$, leaf capacity $\mathcal{L}_{\text{capacity}}$

**Ensure:** a Balanced quadtree with spatial subdivision

1: **function** INITIALISEQUADTREE($R, \mathcal{N}_{\text{capacity}}, \mathcal{L}_{\text{capacity}}$)
2:      $Q \leftarrow Q(R(X_{\min}, Y_{\min}, X_{\max}, Y_{\max}), \mathcal{N}_{\text{capacity}}, \mathcal{L}_{\text{capacity}})$
3:      INSERT($p$)
4: **end function**
5: **function** INSERT($p$)
6:      $Q$.insert($P(x_i, y_i, A_i)$)
7:      **if** $\mathcal{N}_{\text{current}}$ is leaf and $|\mathcal{I}(N)| \geq \mathcal{N}_{\text{capacity}}$ **then**
8:          SUBDIVIDE(self)
9:      **end if**
10: **end function**
11: **function** SUBDIVIDE(self)
12:      Add children: $\mathcal{C}(\mathcal{N}_{\text{parent}}) \leftarrow \{Q_{\text{NW}}, Q_{\text{NE}}, Q_{\text{SW}}, Q_{\text{SE}}\}$
13:      Distribute points to children nodes
14:      **for** each $p \in \mathcal{I}(\mathcal{N}_{\text{parent}})$ **do**
15:          **for** each $\mathcal{N} \in \mathcal{C}(\mathcal{N}_{\text{parent}})$ **do**
16:              **if** CONTAINS($R(\mathcal{N}), X_p, Y_p$) **then**
17:                  INSERT($\mathcal{N}, p$)
18:              **end if**
19:          **end for**
20:      **end for**
21: **end function**
22: **begin**
23: INITIALISEQUADTREE($R, \mathcal{N}_{\text{capacity}}, \mathcal{L}_{\text{capacity}}$)
24: **end**

---

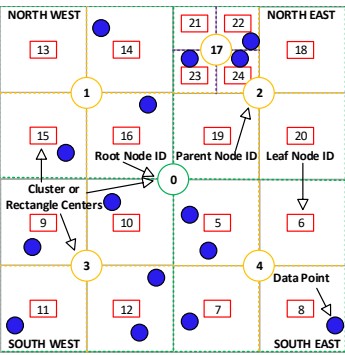

Figure 8: Recursive Quadtree-Based Spatial Segmentation.

efficiency and preventing excessive granularity. This initialisation provides a scalable foundation for dynamic spatial segmentation.

**Step 2: Data Insertion:** Each crime incident is represented in 2D space as $P(x, y, \mathbf{A})$, where $(x, y)$ are the spatial coordinates and $\mathbf{A}_i$ contains associated attributes (e.g., timestamp). Data points are sequentially inserted into the quadtree using $Q$.insert($P(x_i, y_i, \mathbf{A}_i)$), ensuring the spatial and attribute data are efficiently stored in the hierarchical structure.

*Handling Data Points:* For points near node boundaries, the quadtree validates placement using rectangle intersection checks to avoid misplacement or data loss. Points shared across multiple regions are assigned accurately by verifying spatial overlap. Intersection conditions are defined as: $X_{\max} \geq X'_{\min}$ and $X_{\min} \leq X'_{\max}$ and $Y_{\max} \geq Y'_{\min}$ and $Y_{\min} \leq Y'_{\max}$. This ensures precise handling of boundary data, particularly for points spanning multiple regions.

*Capacity Constraints and Leaf Nodes:* A leaf node ($\mathcal{N}_{\text{leaf}}$) is a terminal node in the quadtree that stores localized data points. If the number of interim data points ($\mathcal{I}(N)$) in a leaf node is below the $\mathcal{N}_{\text{capacity}}$, a new data point ($p$) can be directly appended to the set of interim points in that node. Interim data points serve as a temporary storage mechanism for localised data within a node. When $|\mathcal{I}(N)| \geq \mathcal{N}_{\text{capacity}}$, the node undergoes recursive subdivision, dynamically partitioning the space to adapt to data density and ensure efficient spatial representation.

**Step 3: Recursive Subdivision:** When the leaf node ($\mathcal{N}_{\text{leaf}}$) exceeds its capacity ($\mathcal{N}_{\text{capacity}}$), it transitions into a parent node ($\mathcal{N}_{\text{parent}}$) to dynamically portion the spatial (crime) environment and better capture local crime density variations. *Creating child Nodes:* The parent node is subdivided into four child nodes: Northwest (NW), Northeast (NE), Southwest (SW), and Southeast (SE). The boundaries for each child node are calculated using the parent's midpoint coordinates:

$$X_{\text{mid}} = \frac{X_{\text{min}} + X_{\text{max}}}{2} \quad \text{and} \quad Y_{\text{mid}} = \frac{Y_{\text{min}} + Y_{\text{max}}}{2}.$$

Each child node ($Q_i$) is initialized as: $Q_i = Q(R_i, \mathcal{N}_{\text{capacity}}, \mathcal{L}_{\text{capacity}})$, ($i \in \{\text{NW}, \text{NE}, \text{SW}, \text{SE}\}$), where $R_i$ represents the spatial boundaries of the $i - th$ quadrant. This ensures a finer spatial granularity, enabling each child node to focus on localised crime patterns.

*Node Hierarchy and Depth Assignment:* Each child node is assigned a unique identifier, and its depth level is based on the parent. This can be expressed as:

$$\mathcal{C}(\mathcal{N}_{\text{parent}}) = \mathcal{C}(\mathcal{N}_{\text{parent}}) \cup \{N_{\text{NW}}, N_{\text{NE}}, N_{\text{SW}}, N_{\text{SE}}\}.$$

The newly created child nodes are appended to the parent node's children list, forming a hierarchical structure with refined spatial granularity.

*Data Redistribution:* After the child nodes ($N_{\text{NW}}, N_{\text{NE}}, N_{\text{SW}}, N_{\text{SE}}$) are created, data points from the parent node ($\mathcal{N}_{\text{parent}}$) are redistributed to the appropriate child nodes ($N_{\text{child}}$) based on their spatial coordinates. A data point $p \in I(N_{\text{parent}})$ is assigned to a $N_{\text{child}}$ if its coordinates ($X_p, Y_p$) lie within the spatial boundaries $R(N_{\text{child}})$:

$$\forall p \in I(N_{\text{parent}}), \; p \rightarrow N_{\text{child}} \text{ if } (X_p, Y_p) \in R(N_{\text{child}}).$$

To optimize memory usage and computational efficiency, QREPM uses references (or pointers) for redistribution rather than duplicating the data points. Importantly, the parent node retains its original data points while referencing child nodes.

*Recursive Process:* The subdivision process continues recursively until all nodes satisfy the capacity constraint ($|\mathcal{I}(N)| \leq \mathcal{N}_{\text{capacity}}$), ensuring that each leaf node contains a manageable number of data points. This dynamic adaptation allows the quadtree to efficiently handle varying data densities across the study area.

A.2 ADAPTIVE SUBDIVISION THRESHOLD WITH SMART-QT HYPERPARAMETERS.

**Adaptive Subdivision Threshold** ($T_{\text{max}}$)**:** Instead of using a fixed threshold $M$, SMART-QT dynamically determines the subdivision condition using a variance-aware density function. Specifically, the node capacity $T_{\text{max}}$ is computed as:

$$T_{\text{max}} \quad = \quad \max\Big(T_{\text{min}}, \; \min\Big(\frac{\beta}{1 + \frac{\sigma^2}{\gamma}} + \frac{|P|}{\delta}, \; T_{\text{cap}}\Big)\Big),$$

In Eq. (1), $\sigma^2$ denotes the variance of crime counts within node $P$, and $|P|$ is the number of points in that node. The terms $T_{\text{min}}$ and $T_{\text{cap}}$ serve as lower and upper bounds, respectively, computed by scaling the reference parameters $\alpha$ and $\kappa$ according to the dataset size. The parameter $\beta$ controls the inverse relationship to local variance; a higher $\beta$ emphasises variance reduction, while $\gamma$ modulates sensitivity to this variance.

The final term, involving $\delta$, adds a density-dependent penalty proportional to the current node size $|P|$, encouraging more conservative subdivision in already dense regions. However, since the threshold is computed based on the density of the node, we constrain it using fixed caps: $T_{\text{min}}$ ensures

---

**Algorithm 2** SMART Quadtree Construction

---

**Require:** Dataset $\mathcal{D} = \{(\mathbf{x}_i, t_i, c_i)\}_{i=1}^N$, reference size $n_{\text{ref}}$
**Ensure:** SMART-QT structure with hierarchical partitioning
1: Compute scaling factor $s \leftarrow N/n_{\text{ref}}$
2: Scale parameters: $\alpha, \beta, \kappa, \lambda, \gamma, \delta, \texttt{min\_base}$
3: Extract spatio-temporal features $\mathbf{x}_i$ from $(x_i, y_i, t_i)$
4: Initialize root node $\mathcal{N}_0$ with boundary $\mathcal{S}$, data $\mathcal{D}_0 = \mathcal{D}$, depth $d = 0$
5: **function** BUILDTREE($\mathcal{N}_j, d$)
6:     $n_j \leftarrow |\mathcal{D}_j|$                                          ▷ Point count in node
7:     $\sigma_j^2 \leftarrow \text{Var}(\{c_i \in \mathcal{D}_j\})$                         ▷ Crime count variance
8:     Compute $T_{\max} \leftarrow f(\alpha, \beta, \kappa, n_j, \sigma_j^2)$        ▷ Adaptive capacity
9:     Compute $L_{\max} \leftarrow g(\lambda, \gamma, \delta, N)$            ▷ Adaptive depth limit
10:     **if** $d < L_{\max}$ **and** $n_j \geq T_{\max}$ **then**
11:         **if** $d \bmod \nu = 0$ **then**          ▷ $\nu = 3$ if $N > 10^6$, else 1
12:             Recompute $T_{\max}, L_{\max}$        ▷ Adapt to local density
13:         **end if**
14:         $(x_{\text{med}}, y_{\text{med}}) \leftarrow \text{Median}(\mathcal{D}_j)$          ▷ Median-based split
15:         Partition $\mathcal{D}_j$ into $\{\mathcal{D}_{NW}, \mathcal{D}_{NE}, \mathcal{D}_{SW}, \mathcal{D}_{SE}\}$
16:         Create child nodes $\{\mathcal{N}_{NW}, \mathcal{N}_{NE}, \mathcal{N}_{SW}, \mathcal{N}_{SE}\}$
17:         Propagate parent features to children
18:         **for** each child $\mathcal{N}_k \in \{\mathcal{N}_{NW}, \mathcal{N}_{NE}, \mathcal{N}_{SW}, \mathcal{N}_{SE}\}$ **do**
19:             BUILDTREE($\mathcal{N}_k, d + 1$)
20:         **end for**
21:     **end if**
22: **end function**
23: BUILDTREE($\mathcal{N}_0, 0$)
24: MERGESMALLLEAVES                       ▷ Algorithm 2 in Appendix
25: **return** Final SMART-QT tree

---

the threshold does not fall below a minimum subdivision size, and $T_{\text{cap}}$ ensures it does not grow unbounded in low-variance or high-density regions. This clipping mechanism ensures robustness and consistency across varying conditions. Together, these scaled constants ensure that node capacities dynamically reflect both global dataset characteristics and local statistical heterogeneity.

These are all auto-tunable parameters that adapt based on the dataset size. This formulation ensures that nodes with high variance (i.e., potential hotspots) are subdivided more aggressively, while more homogeneous regions retain larger spatial coverage. In contrast to fixed heuristics, this adaptive mechanism is sensitive to both local and global dataset properties. All parameter values used in the adaptive threshold equations, including $\alpha, \beta, \gamma, \delta, \kappa$, and $\lambda$, are dynamically scaled based on dataset size to ensure consistent behavior across varying data volumes; see Dynamic Parameter Scaling for details.

**SMART-QT Hyperparameters:** Auto-tuneable hyper-parameters used for the SMART-QT are:

Table 8: SMART-QT Hyperparameters and Their Roles

| Parameter | Description |
|---|---|
| $\alpha$ | Minimum allowable node capacity to prevent over-splitting. |
| $\beta$ | Controls inverse relationship between variance and node capacity. |
| $\gamma$ | Stabilizes variance sensitivity in the $T_{\max}$ formula. |
| $\delta$ | Penalizes node size when computing adaptive capacity. |
| $\kappa$ | Upper cap for node capacity, computed as $\kappa = \frac{n_{\text{total}}}{\lambda}$. |
| $\lambda$ | Global scaling factor used in defining $\kappa$. |
| $\eta$ | Multiplier controlling the maximum depth of the tree. |
| $\nu$ | Interval for periodic threshold re-tuning at node depth levels. |

## A.3 STRATEGIC SMALL-LEAF MERGING DETAILS

To prevent over-fragmentation in sparse, low-density, or noisy regions, SMART-QT integrates a variance-aware *Strategic Small-Leaf Merging* mechanism. Classical quadtrees often split uniformly, producing many tiny leaves in areas that do not contain sufficient information for reliable local modelling. Such over-partitioning leads to poor generalisation, unnecessary computational overhead, and unstable models in underrepresented regions. The merging strategy addresses these shortcomings by consolidating weak or noisy leaves into statistically coherent siblings.

**Formal Definition:** Let a node $\mathcal{N}$ contain a point set $P$ within spatial area $A$, and define its density as $\rho = \frac{|P|}{A}$. Two categories of nodes are flagged as merge candidates: (i) **Small nodes:** nodes with $0 < |P| < \tau$, where $\tau$ is a user-defined minimum threshold for reliable modelling. (ii) **Density outliers:** nodes whose density deviates significantly from the global distribution across all leaves. Specifically, a node is flagged if

$$\rho < Q_1 - \phi \cdot \text{IQR} \quad \text{or} \quad \rho > Q_3 + \phi \cdot \text{IQR},$$

where $Q_1$ and $Q_3$ denote the first and third quartiles of the density distribution, $\text{IQR} = Q_3 - Q_1$, and $\phi$ is a tunable sensitivity parameter (default $\phi = 1.5$).

**Merging Criteria:** For each candidate node $\mathcal{N}_c$, the algorithm searches for a *compatible sibling* $\mathcal{N}_s$, defined as a leaf node under the same parent that: (i) Satisfies the safe upper bound: $|P_c| + |P_s| < \tau_{\max}$, where $\tau_{\max} = 2.5 \cdot \tau$. (ii) Minimises density mismatch: $\mathcal{N}_s = \arg\min_s |\rho_c - \rho_s|$. If multiple siblings satisfy the criteria, the one with closest density is chosen. All points from $\mathcal{N}_c$ are then reassigned to $\mathcal{N}_s$, and $\mathcal{N}_c$ is marked inactive. The merged relationship is stored in a mapping: `merged_pairs`$[c] = s$, where $c$ and $s$ are node IDs of the candidate and sibling.

---

**Algorithm 3** Strategic Small-Leaf Merging

---

**Require:** Leaf set $\mathcal{L}$, threshold $\tau$, sensitivity $\phi = 1.5$
**Ensure:** Updated quadtree with merged nodes, mapping `merged_pairs`
 1: Compute density statistics: $Q_1, Q_3, \text{IQR}$ for all $\mathcal{N} \in \mathcal{L}$
 2: $\mathcal{C} \leftarrow \{\mathcal{N} \in \mathcal{L} \mid (0 < |P| < \tau) \vee (\rho < Q_1 - \phi \cdot \text{IQR}) \vee (\rho > Q_3 + \phi \cdot \text{IQR})\}$
 3: **for** each candidate $\mathcal{N}_c \in \mathcal{C}$ **do**
 4: $\quad S \leftarrow \{\text{siblings of } \mathcal{N}_c \text{ under same parent}\}$
 5: $\quad S_{\text{valid}} \leftarrow \{\mathcal{N}_s \in S \mid |P_c| + |P_s| < 2.5\tau\}$
 6: $\quad$ **if** $S_{\text{valid}} \neq \emptyset$ **then**
 7: $\quad\quad \mathcal{N}_s \leftarrow \arg\min_{\mathcal{N}_s \in S_{\text{valid}}} |\rho_c - \rho_s|$
 8: $\quad\quad P_s \leftarrow P_s \cup P_c$ $\qquad\qquad\qquad\qquad\qquad$ ▷ Merge point sets
 9: $\quad\quad$ Mark $\mathcal{N}_c$ as inactive
10: $\quad\quad$ `merged_pairs`$[c] \leftarrow s$
11: $\quad$ **end if**
12: **end for**
13: **return** Updated leaf set, merged pairs mapping

---

**Geometric Example:** Merging redefines spatial boundaries while preserving coverage. For adjacent nodes: $\mathcal{N}_1 : [x_1, x_m] \times [y_1, y_2]$, $\mathcal{N}_2 : [x_m, x_2] \times [y_1, y_2]$, the merged node spans the union: $\mathcal{N}_{\text{merged}} : [x_1, x_2] \times [y_1, y_2]$. The merged region inherits the combined point set $P_1 \cup P_2$ and aggregate statistics while maintaining spatial continuity.

**Discussion:** This procedure ensures that sparse, noisy, or outlier leaves do not distort training. Instead, they are merged into nearby statistically compatible siblings, maintaining spatial coherence and improving stability of local models. The result is a structurally simplified quadtree that reduces overfitting, lowers computational cost, and yields more robust predictions in real-world heterogeneous datasets such as crime logs.

## A.4 PERIODIC THRESHOLD RE-TUNING:

The re-tuning frequency $\nu$ controls the trade-off between adaptability and efficiency. For instance:

- $\nu = 1$: Re-tuning at every level (maximal adaptability, higher cost).

- $\nu = 3$: Re-tuning every third level (improved efficiency for very large datasets).

The optimal $\nu$ is selected empirically based on dataset size and desired spatial resolution. When triggered, the values of $T_{\max}$ and $L_{\max}$ are recomputed using the same variance-aware formulations introduced earlier in Equation (1) and Equation (2). Specifically, the local variance $\sigma^2_{\text{local}}$ is computed over the node's crime counts, and the re-tuned thresholds are propagated recursively to its child nodes. This local adjustment ensures that the quadtree structure dynamically adapts to regional crime patterns—splitting more aggressively in high-variance urban centres, while allowing larger aggregated regions in more uniform or low-density areas. SMART-QT maintains a balance between spatial granularity and computational efficiency by tailoring its partitioning behaviour at multiple depths.

## A.5 FEATURE CONSTRUCTION AND PROPAGATION (IMPLEMENTATION DETAILS):

We derive a standardised spatio-temporal feature vector $\mathbf{x}_i$ for each data point and retain it (by reference) at every node so that parent and child nodes share identical input features plus the parent prediction. Preprocessing: raw timestamps $t_i$ are converted to UNIX seconds and used to sort and align time series; any missing temporal fields are inferred from the timestamp. Target counts $c_i$ are transformed with $\log(1 + c_i)$ and then scaled to $[0, 1]$ using a MinMax scaler fitted on training data. Feature standardisation: continuous features (coordinates, rolling statistics, lags) are standardised to zero mean and unit variance using a StandardScaler fitted on the training set; categorical/binary features (is_weekend, is_holiday) are left as 0/1. Lags and rolling statistics: we compute lagged counts $c_{i-k}$ for $k \in \{1, 2, 3\}$ as $c_{i-k} = c(t_i - k \cdot \Delta)$ where $\Delta$ is one day (daily aggregation); gaps (missing days) are handled by imputing with the node median count. The 7-day rolling mean is computed as $\text{roll7}_i = \frac{1}{7} \sum_{j=1}^{7} c_{i-j}$ (using the same gap policy); for short time windows at series start we use the available history (no padding). Cyclic encodings use standard sin/cos transforms: $\text{hour}_{\sin} = \sin(2\pi \cdot \text{hour}/24)$, $\text{hour}_{\cos} = \cos(2\pi \cdot \text{hour}/24)$, and similarly for month with period 12. The full propagated feature set (kept for every point and passed from parent→child) is: {scaled longitude, scaled latitude, UNIX date, hour, day_of_week, is_weekend, is_holiday, day_of_month, month, year, $\text{hour}_{\sin}$, $\text{hour}_{\cos}$, $\text{month}_{\sin}$, $\text{month}_{\cos}$, lag1, lag2, lag3, roll7, seasonal flags (spring/summer/fall/winter)}. In addition, we propagate a dedicated 'Prediction' column containing the parent model's prediction for each point; children append this value to their input vector and warm-start weights from parent models (XGBoost booster or NN weights). All derived feature computation code, imputation policy and scaler objects are saved with the training artifacts to ensure exact reproducibility.

## A.6 DYNAMIC PARAMETER SCALING — RATIONALE AND IMPLEMENTATION

To make SMART-QT robust across datasets of very different scales (from $10^5$ to $10^7$ points), we dynamically scale a set of tuned base constants by dataset size. The scaling factor is defined as

$$s = \frac{n_{\text{total}}}{n_{\text{ref}}},$$

where $n_{\text{total}}$ is the dataset size and $n_{\text{ref}}$ is a fixed reference (in our experiments $n_{\text{ref}} = 10^6$). This ensures that node capacity thresholds and depth limits grow smoothly with dataset size, preserving both scalability and stability.

**Clipping rationale:** Two practical constraints motivate the clipping ranges used (e.g. $\lambda \in [5, 20]$, $\gamma, \delta \in [1, 5]$): (i) **Numerical stability and interpretability:** Parameters such as $\lambda$ control exponential/logarithmic behaviours in threshold updates; letting them grow unbounded causes sudden jumps, while values too close to zero make the method insensitive to variance. (ii) **Computational control:** Clipping keeps effective complexity within practical bounds so that quadtree growth, re-tuning frequency, and sibling-merge heuristics remain efficient on real hardware. Empirically, these ranges strike a balance across city datasets from $\approx 10^5$ to $\approx 10^7$ records.

**Implementation notes:** We distinguish two parameter groups: (i) **Integer-scale parameters** ($\alpha, \beta, \kappa, \texttt{min\_base}$): scaled linearly with $s$, then rounded. (ii) **Smooth parameters** ($\lambda, \gamma, \delta$): scaled with $\log(1 + s)$ and clipped.

**Algorithm sketch:**

```
Inputs:
  N       # dataset size
  N_ref   # reference dataset size (1e6)
  base = {alpha=2000, kappa=50000, min_base=5000,
          beta=50000, lambda_val=10, gamma=2, delta=2}

Compute scaling factor:
  s = max(1.0, N / N_ref)
  log_scale = 1 + log(s)

Scale parameters:
  For alpha, beta, kappa, min_base:
      scaled[param] = max(1, round(base[param] * s))
  For lambda, gamma, delta:
      scaled[param] = clip(round(base[param] * log_scale),
                           lower_bound, upper_bound)
```

**Role of `min_base`, $T_{\min}$, $T_{\max}$ and $T_{\text{cap}}$:** We recall the adaptive node capacity formula from Eq. 1:

$$T_{\max} = \max\Big(T_{\min},\ \min\Big(\frac{\beta}{1 + \sigma^2/\gamma} + \frac{|P|}{\delta},\ T_{\text{cap}}\Big)\Big),$$

where $\sigma^2$ is local variance of crime counts and $|P|$ is the number of points in the node.

- $T_{\min}$ (set by `min_base`) prevents thresholds from falling too low in homogeneous or noisy regions.
- $T_{\text{cap}}$ (scaled from $\kappa$) prevents thresholds from growing without bound in low-variance dense areas.
- The first term $\frac{\beta}{1+\sigma^2/\gamma}$ decreases as local variance grows, forcing earlier splits in hotspots.
- The second term $\frac{|P|}{\delta}$ prevents unnecessary splitting of already dense nodes.

**Worked numeric examples:** For illustration, consider defaults $\beta = 50{,}000$, $T_{\min} = 5{,}000$, $T_{\text{cap}} = 50{,}000$, $\gamma_0 = 2$, $\delta_0 = 2$, with $n_{\text{ref}} = 10^6$ and dataset size $n_{\text{total}} = 5 \times 10^6$ ($s = 5$).

Then:
$$\log(1 + s) = \log(6) \approx 1.7918, \quad \gamma \approx 1.79, \quad \delta \approx 3.58.$$

- **Case A (low variance, $\sigma^2 = 0.5$, $|P| = 1{,}000$):**
  $\frac{\beta}{1+\sigma^2/\gamma} \approx 39{,}073.3$, $\frac{|P|}{\delta} \approx 279.1 \Rightarrow T_{\max} \approx 39{,}352.3$.
- **Case B (medium variance, $\sigma^2 = 5$, $|P| = 5{,}000$):**
  $\frac{\beta}{1+5/\gamma} \approx 13{,}193.9$, $\frac{|P|}{\delta} \approx 1{,}395.3 \Rightarrow T_{\max} \approx 14{,}589.2$.
- **Case C (high variance, $\sigma^2 = 50$, $|P| = 20{,}000$):**
  $\frac{\beta}{1+50/\gamma} \approx 1{,}729.5$, $\frac{|P|}{\delta} \approx 5{,}581.1 \Rightarrow T_{\max} \approx 7{,}310.5$.

**Interpretation:** As local variance increases, the $\beta$-term shrinks and $T_{\max}$ falls, enabling more aggressive splitting in hotspots. The density penalty term moderates this behaviour in already dense nodes. On small datasets ($s < 1$), $\gamma$ and $\delta$ clip near 1, which makes $|P|/\delta$ larger, so $T_{\max}$ is higher (more conservative splitting). Thus, the rule:

- is monotone in both $n_{\text{total}}$ and $\sigma^2$,
- is provably bounded and asymptotically $\Theta(\log n_{\text{total}})$,
- balances expressiveness (deep trees where variance is high) with efficiency (shallower trees in uniform regions).

This variance-aware scaling ensures SMART-QT remains adaptive, statistically reliable, and computationally efficient across a wide range of dataset scales.

# B CARE PREDICTIVE MODEL

## B.1 CARE-PM ALGORITHM

---

**Algorithm 4** CARE Predictive Model Framework

---

**Require:** Trained SMART-QT structure, training data $\mathcal{D}_{\text{train}} = \{(\mathbf{x}_i, y_i)\}_{i=1}^N$, test data $\mathbf{x}_{\text{test}}$
**Ensure:** Predictions $\hat{y}_{\text{final}}$ for test data
1: Train baseline model $f_{\text{base}}$ on $\mathcal{D}_{\text{train}}$ to get $\hat{y}_0(i) = f_{\text{base}}(\mathbf{x}_i)$
2: Standardize $\hat{y}_0$ and augment features: $\mathbf{x}_{i,\text{aug}} = [\mathbf{x}_i, \hat{y}'_0]$
3: Train root model $f_{\text{root}}$ on $\mathbf{x}_{i,\text{aug}}$, yielding $\hat{y}_1(i) = f_{\text{root}}(\mathbf{x}_{i,\text{aug}})$
4: **function** TRAINNODE$(\nu, \mathcal{D}_\nu, f_{\pi(\nu)})$
5:     **if** $|\mathcal{D}_\nu| \geq \tau$ and $\nu$ not merged **then**
6:         Initialize $f_\nu$ with $f_{\pi(\nu)}$'s parameters (e.g., booster trees)
7:         Train $f_\nu$ on $\mathcal{D}_\nu$ with $\hat{y}'_{\pi(\nu)}$, augmenting $\mathbf{x}_{i,\text{aug},\nu}$
8:         Update predictions: $\hat{y}_\nu(i) = f_\nu(\mathbf{x}_{i,\text{aug},\nu})$
9:     **else**
10:         Retain $\hat{y}_{\pi(\nu)}(i)$ for skipped/merged nodes
11:     **end if**
12: **end function**
13: **for** each node $\nu$ in breadth-first order **do**
14:     $\mathcal{D}_\nu \leftarrow$ points in $\nu$'s region
15:     TRAINNODE$(\nu, \mathcal{D}_\nu, f_{\pi(\nu)})$
16: **end for**
17: **function** PREDICT$(\mathbf{x}_{\text{test}})$
18:     Compute $\hat{y}_0 = f_{\text{base}}(\mathbf{x}_{\text{test}})$, augment to $\mathbf{x}_{\text{test,aug}}$
19:     Set $\hat{y}_k \leftarrow f_{\text{root}}(\mathbf{x}_{\text{test,aug}})$
20:     **for** each level $k$ from 1 to leaf **do**
21:         Route $\mathbf{x}_{\text{test}}$ to child node $\nu$
22:         Standardize $\hat{y}_k$ to $\hat{y}'_k$, augment $\mathbf{x}_{\text{test,aug},\nu}$
23:         $\hat{y}_{k+1} \leftarrow f_\nu(\mathbf{x}_{\text{test,aug},\nu})$
24:     **end for**
25:     **return** $\hat{y}_{\text{final}} = \hat{y}_{\text{last}}$
26: **end function**
27: **return** final SMART-QT tree with all node models $f_\nu$
28: **return** $\hat{y}_{\text{final}}$ for $\mathbf{x}_{\text{test}}$

---

## B.2 DETAILED ROOT NODE TRAINING

The CARE Predictive Model begins by training the *root node*, which covers the entire spatial domain of the dataset. Because the root has no parent, its purpose is to establish a global baseline that captures overall spatio-temporal crime dynamics before finer regional refinements are learned in deeper quadtree nodes. Below, we expand on the compact description given in the main text.

**Step 1: Baseline Global Model.** We first train a global baseline regressor $f_{\text{base}}$ on the full training set $\mathcal{D}_{\text{train}} = \{(\mathbf{x}_i, y_i)\}_{i=1}^N$, where $\mathbf{x}_i \in \mathbb{R}^d$ is the spatio-temporal feature vector and $y_i$ is the observed crime count. This model minimises the mean squared error (MSE):

$$\theta^\star_{\text{base}} = \arg\min_\theta \frac{1}{|\mathcal{D}_{\text{train}}|} \sum_{i=1}^{|\mathcal{D}_{\text{train}}|} \left(y_i - f_{\text{base}}(\mathbf{x}_i; \theta)\right)^2.$$

In our experiments, $f_{\text{base}}$ is instantiated as either: (i) an XGBoost regressor with depth $= 6$, learning rate $= 0.1$, and 200 trees; or (ii) a two-layer neural network with hidden size $[64, 32]$ and ReLU activations. This baseline produces preliminary estimates $\hat{y}_{\text{base}}(i) = f_{\text{base}}(\mathbf{x}_i)$ for each input.

**Step 2: Standardisation of Predictions.** To ensure comparability across nodes and prevent numerical dominance of the baseline predictions, we standardise them to zero mean and unit variance:

$$\hat{y}'_{\text{base}}(i) = \frac{\hat{y}_{\text{base}}(i) - \mu_{\hat{y}_{\text{base}}}}{\sigma_{\hat{y}_{\text{base}}}},$$

where $\mu_{\hat{y}_{\text{base}}} = \frac{1}{N}\sum_{i=1}^{N}\hat{y}_{\text{base}}(i)$ and $\sigma_{\hat{y}_{\text{base}}} = \sqrt{\frac{1}{N}\sum_{i=1}^{N}\big(\hat{y}_{\text{base}}(i) - \mu_{\hat{y}_{\text{base}}}\big)^2}$. This step is essential: it prevents inflated variance in downstream models and ensures the "Prediction" feature has the same scale as the other spatio-temporal features.

**Step 3: Feature Augmentation.** The standardised predictions are appended as an additional feature, yielding the augmented input:

$$\mathbf{x}_{i,\text{aug}} = [\mathbf{x}_i, \hat{y}'_{\text{base}}(i)].$$

This feature encodes the global crime trend for each sample and allows the root node to refine the baseline estimates.

**Step 4: Root Model Training.** The final root model $f_{\text{root}}$ is trained on the augmented dataset $\{(\mathbf{x}_{i,\text{aug}}, y_i)\}_{i=1}^{N}$, minimising the MSE:

$$\theta^{\star}_{\text{root}} = \arg\min_{\theta}\ \frac{1}{N}\sum_{i=1}^{N}\Big(y_i - f_{\text{root}}(\mathbf{x}_{i,\text{aug}}; \theta)\Big)^2.$$

In the tree-based case, $f_{\text{root}}$ is an XGBoost model warm-started from $f_{\text{base}}$. In the neural case, $f_{\text{root}}$ is initialised with the parameters of $f_{\text{base}}$ and fine-tuned on the augmented data. This warm-starting accelerates convergence and ensures that $f_{\text{root}}$ inherits global structure while learning local refinements.

**Step 5: Output.** The root model generates refined predictions:

$$\hat{y}_1(i) = f_{\text{root}}(\mathbf{x}_{i,\text{aug}}),$$

which serve as the top-level forecasts. These are propagated to child nodes during subsequent hierarchical training (see Appendix B.3), ensuring that each level inherits both the global baseline and the root's refined adjustments.

**Implementation Note.** For reproducibility, we set the reference dataset size $n_{\text{ref}} = 10^6$ when computing scaling constants, learning rate decay with depth factor $\gamma = 0.5$, and minimum node size $N_{\min} = 50$. In practice, the baseline and root are implemented as *separate models*: $f_{\text{base}}$ provides a normalised trend feature, while $f_{\text{root}}$ produces the actual top-level prediction.

### B.3 DETAILED CHILD NODE TRAINING

After the root node has established a global baseline, CARE progressively trains the child nodes to refine predictions for local spatial regions. This process is recursive, moving top-down through the SMART-QT hierarchy. We expand here on the compact description given in the main text.

**Step 1: Data Partitioning.** Each child node $\nu$ is associated with a subset of training points $\mathcal{D}_{\nu} = \{(\mathbf{x}_i, y_i)\}_{i \in I_{\nu}}$ that fall within the spatial boundaries of the SMART-QT partition. By construction, these subsets are disjoint across siblings but collectively cover the parent's domain.

**Step 2: Parent Prediction Feature.** For every sample $i \in \mathcal{D}_{\nu}$, we compute the prediction of the parent model $f_{\pi(\nu)}$:

$$\hat{y}_{\pi(\nu)}(i) = f_{\pi(\nu)}(\mathbf{x}_{i,\text{aug}}).$$

Since the magnitude of parent predictions may vary across nodes, we normalise them within $\mathcal{D}_{\nu}$:

$$\hat{y}'_{\pi(\nu)}(i) = \frac{\hat{y}_{\pi(\nu)}(i) - \mu_{\hat{y}_{\pi(\nu)}}}{\sigma_{\hat{y}_{\pi(\nu)}}},$$

where $\mu_{\hat{y}_{\pi(\nu)}}$ and $\sigma_{\hat{y}_{\pi(\nu)}}$ are the mean and standard deviation of $\hat{y}_{\pi(\nu)}$ over $\mathcal{D}_{\nu}$. This ensures scale consistency and prevents numerical drift as predictions are propagated downward.

**Step 3: Feature Augmentation.** The standardised parent prediction is appended to the original feature vector, producing:

$$\mathbf{x}_{i,\text{aug},\nu} = [\mathbf{x}_i, \hat{y}'_{\pi(\nu)}(i)].$$

This augments the child's training input with both local spatio-temporal context ($\mathbf{x}_i$) and inherited parent knowledge.

**Step 4: Model Initialisation.** To accelerate convergence and stabilise learning, each child's regressor $f_\nu$ is *warm-started* from its parent model $f_{\pi(\nu)}$:

- **Tree-based case (XGBoost):** The child booster inherits the parent's ensemble and continues training with additional trees, typically with a reduced learning rate or capped number of new trees $T_{\text{new}}$.

- **Neural case:** The child network is initialised with the parent's weights $\theta_{\pi(\nu)}$ and fine-tuned on $\mathcal{D}_\nu$ with a local learning rate schedule.

**Step 5: Local Objective.** The child model minimises the local mean squared error:

$$\theta_\nu^\star = \arg\min_\theta \; \frac{1}{|\mathcal{D}_\nu|} \sum_{i \in \mathcal{D}_\nu} \left(y_i - f_\nu(\mathbf{x}_{i,\text{aug},\nu}; \theta)\right)^2 \; + \; \lambda_\nu \, \Omega(\theta, \theta_{\pi(\nu)}),$$

where $\Omega(\cdot)$ is a proximity regulariser. For neural models, $\Omega(\theta, \theta_{\pi(\nu)}) = \|\theta - \theta_{\pi(\nu)}\|_2^2$, enforcing similarity to the parent's weights. For XGBoost, the proximity is implicit via warm-starting and limiting additional trees.

**Step 6: Recursive Refinement.** The predictions from each child are then propagated further down to their descendants, repeating Steps 2–5. This recursive refinement continues until leaf nodes are reached, where the most specialised models provide fine-grained predictions for local regions while maintaining coherence with the global structure propagated from the root.

**Implementation Notes.** In practice, small nodes with $|\mathcal{D}_\nu| < N_{\text{min}}$ are exempt from retraining; instead, they reuse their parent's predictions. This avoids overfitting in data-sparse regions. We used $N_{\text{min}} = 50$ in our experiments.

### B.4 FINE-TUNING HYPERPARAMETERS FOR NEURAL NETWORK MODELS

The hyperparameters in Table 6 were selected to balance model complexity with the hierarchical quadtree fine-tuning strategy. At the root level, higher learning rates encourage global pattern discovery, while lower child learning rates stabilize local refinements and prevent catastrophic forgetting. Dropout values scale with model capacity, being higher for Light Transformers and BiLSTMs, moderate for LSTMs, and lower for GRUs and MLPs, reflecting their susceptibility to overfitting. Batch sizes and epoch counts are adjusted according to model complexity and local data availability, with smaller batches and fewer epochs for memory-intensive models (Transformers, BiLSTMs) and larger batches for lighter models (GRUs, MLPs). These choices ensure that each model can refine inherited weights at the leaf nodes effectively without overfitting, maintaining a balance between global generalization at the root and robust local specialization at the leaves.

The learning-rate schedules in Table 7 complement this framework by further stabilizing training and enhancing generalization. Light Transformers utilize a short linear warm-up followed by cosine annealing to gradually reduce the learning rate across child epochs, preventing destabilization of pre-trained attention patterns. RNN models (GRU, LSTM, BiLSTM) employ step decay or ReduceL-ROnPlateau strategies to adaptively reduce learning rates when validation performance plateaus, which is critical for small, sparse quadtree segments. Child-level learning rates are conservatively lower than root-level rates to refine inherited weights without overwriting global patterns. Minimum learning-rate floors, along with weight decay and dropout, further regularize the models. Overall, varying hyperparameters across models is intentional to match their computational requirements and overfitting tendencies, ensuring each model achieves its optimal performance and enabling a fair comparison of prediction accuracy across heterogeneous quadtree segments.

### B.5 Efficient Training and Inference: Complexity Analysis and Empirical Timings

**Per-node complexity.** At inference, each sample traverses at most one model per quadtree level. For an XGBoost regressor with $k$ trees and $p$ features, the per-node cost is $O(p \cdot k)$, since each tree requires at most $p$ feature checks. For a neural network with $H$ layers and widths $\{d_\ell\}$, the per-node cost is $O\!\left(\sum_{\ell=1}^{H} d_{\ell-1} d_\ell\right)$, dominated by dense matrix multiplications. In both cases, $p$, $k$, and $\{d_\ell\}$ are fixed after model tuning, so the per-node cost is constant.

**Depth dependence.** Let $L$ be the quadtree depth and $n$ the dataset size. Each sample evaluation requires $O(L)$ node visits. Since SMART-QT enforces $L = O(\log n)$ (Eq. 2), the total complexity per sample is $O(L) = O(\log n)$.

**Empirical timings.** To validate the constant-factor assumption, we benchmarked prediction time for both model types on a commodity GPU/CPU setup. The average inference time per sample across varying dataset sizes ($n$) is shown below.

| Dataset size ($n$) | Depth ($L$) | XGBoost per-sample (ms) | NN per-sample (ms) |
| --- | --- | --- | --- |
| $10^4$ | 6 | 0.05 | 0.07 |
| $10^5$ | 8 | 0.06 | 0.08 |
| $10^6$ | 12 | 0.08 | 0.11 |

Table 9: Empirical inference times for XGBoost ($k = 200$, $p = 20$) and NN ($H = 3$, widths $\{64, 32, 1\}$). Timings averaged over 10 runs. Per-sample cost grows with depth $L$, consistent with $O(\log n)$ scaling, but remains within practical bounds.

*Implementation note:* For both the XGBoost and NN instantiations of SMART-CARE, we fix the architecture and hyperparameters before training. This ensures that per-node inference cost does not scale with tree depth or dataset size, validating the $O(\log n)$ inference complexity assumption.

**Discussion:** The results confirm that (i) per-node inference cost remains nearly constant once models are fixed, and (ii) per-sample runtime grows only logarithmically with dataset size. This validates the theoretical $O(\log n)$ complexity for both the XGBoost and NN instantiations of SMART-CARE.

### B.6 Hierarchical or Top-Down Inference (Prediction) Procedure:

Once all node models are trained, predicting the outcome for an evaluation data point involves *pre-processing, routing the point through the quadtree and refining the prediction at each level*, mimicking the training sequence. Starting at the root node, the process proceeds as follows: (i) **Baseline Initialization:** For a new input sample $\mathbf{x}$ (with spatio-temporal features), we obtain an initial prediction from the baseline model, $\hat{y}_0 = f_{\text{base}}(\mathbf{x})$. This prediction is standardised (zero-mean, unit-variance) and appended to $\mathbf{x}$, forming the augmented feature vector $\mathbf{x}_{\text{aug}} = [\mathbf{x}, \hat{y}_0']$, consistent with the training phase. (ii) **Root Prediction:** The augmented features are fed into the root model to obtain the first refined prediction: $\hat{y}_1 = f_{\text{root}}(\mathbf{x}_{\text{aug}})$. This $\hat{y}_1$ represents the model's estimate after accounting for global crime patterns, serving as the starting point for further refinement. (iii) **Recursive Refinement:** The sample $\mathbf{x}$ is routed to the child node $\nu$ containing its coordinates, carrying $\hat{y}_1$ as a standardized "Prediction" feature in $\mathbf{x}_{\text{aug},\nu} = [\mathbf{x}, \hat{y}_1']$. The child model computes $\hat{y}_2 = f_\nu(\mathbf{x}_{\text{aug},\nu})$, refining the prediction for the sub-region. This top-down traversal continues recursively: at each node $\nu$, the latest parent prediction $\hat{y}_k$ augments the features, producing $\hat{y}_{k+1} = f_\nu(\mathbf{x}_{\text{aug},\nu})$, and routing to the next child if it exists. For merged or skipped nodes, the parent's prediction is retained, bypassing local updates. Traversal stops at a leaf or when no child model exists, yielding the final prediction $\hat{y}_{\text{final}}$. Formally, the inference path yields a sequence $\hat{y}_0 \to \hat{y}_1 \to \cdots \to \hat{y}_{\leq L_{\max}}$, terminating in the final prediction $\hat{y}_{\text{final}}$.

## C  ABLATION VARIANT:

### C.1  ADAPTIVE MEDIAN-BASED LEAF NODE PREDICTIVE MODEL (AMB-LNPM)

The Adaptive Median-Based Leaf Node Predictive Model (AMB-LNPM) adopts SMART-CARE's adaptive median-based splitting to partition spatial regions, ensuring more balanced data distribution for leaf node predictions. However, it omits SMART-CARE's parent-child knowledge transfer and model refinement, preventing the model from leveraging hierarchical relationships to inform child node predictions or iteratively fine-tune across scales. This absence results in an MAE of 4.18, higher than SMART-CARE's 0.24–0.27, despite relatively balanced splits (variance of points per leaf node: 7820000.00). The lack of hierarchical learning increases memory usage to 40.20 MB, reflecting inefficiencies in processing large datasets like NYPD. AMB-LNPM's performance highlights the importance of hierarchical modelling in capturing complex spatio-temporal crime patterns, as its scalability and accuracy suffer without SMART-CARE's full hierarchical framework.

### C.2  STATIC MID-POINT BASED REFINE ENSEMBLE PREDICTIVE MODEL (SMID-REPM)

The Static Mid-point Based Refine Ensemble Predictive Model (SMID-REPM) employs static mid-point splitting while integrating SMART-CARE's parent-child knowledge transfer and model refinement for leaf node predictions. This hierarchical learning enables child nodes to inherit and fine-tune parent models, improving convergence over non-hierarchical variants. However, SMID-REPM lacks SMART-CARE's adaptive median-based splitting, resulting in uneven data distribution (variance of points per leaf node: 6100000.00). This limitation reduces fine-grained prediction accuracy, yielding an MAE of 3.45, better than SMID-LNPM's 5.62 but worse than SMART-CARE's 0.24–0.27. The static partitioning also increases range query time to 0.0165 seconds, indicating inefficiencies in processing high-density crime regions like Chicago. Consequently, SMID-REPM struggles to balance computational efficiency and prediction accuracy, particularly in datasets with significant spatial variability, limiting its effectiveness compared to SMART-CARE's adaptive approach.

### C.3  STATIC MID-POINT BASED LEAF NODE PREDICTIVE MODEL (SMID-LNPM)

The Static Mid-point Based Leaf Node Predictive Model (SMID-LNPM) constructs a quadtree using fixed midpoint splitting to partition spatial regions, focusing on leaf node predictions without incorporating SMART-CARE's parent-child knowledge transfer or model refinement mechanisms. This absence of hierarchical relationships prevents SMID-LNPM from leveraging parent node models to inform child node predictions, limiting its ability to capture spatial dependencies across scales. Additionally, the lack of refinement hinders iterative model tuning, leading to suboptimal performance in complex crime patterns. Consequently, SMID-LNPM achieves a higher prediction error, with an MAE of 8.62, compared to SMART-CARE's 0.24–0.27, due to static splits causing uneven data distribution (variance of points per leaf node: 6350000.00). This imbalance increases computational overhead, as the model struggles to adapt to high-density regions like NYC, resulting in reduced scalability and accuracy in capturing fine-grained crime trends.

## D  EXPERIMENTS AND RESULTS

### D.1  EXPERIMENTAL SETUP

We use two large-scale crime datasets:

- **NYPD Complaint Data** (7.8M rows, 18 attributes, 2008–2023) Department (2023).
- **Chicago Crime Data** (8.2M rows, 22 attributes, 2001–2024) of Chicago Police (2024).

From each dataset, we extract spatio-temporal features, including `Date-Time`, `Longitude`, and `Latitude`, with temporal attributes engineered (day, week, month, year, cyclic encodings). The SMART-QT structure distributes samples into quadtree nodes based on spatial density, producing balanced training subsets.

For CARE-PM, we employ an XGBoost regressor, tuned via `GridSearchCV` over $n_{\text{estimators}}$ and learning rate. Training proceeds in a breadth-first traversal: parent node predictions are appended

as features and passed to children to enable hierarchical refinement. Hyperparameters are tuned only at the root level and inherited by descendants to reduce search cost. For completeness, we also experimented with a lightweight neural variant (transformer-based regressor) for CARE-PM, using the same parent-to-child inheritance strategy. However, our primary reported results focus on the XGBoost-based implementation.

## D.2 HEATMAP COMPARISON OF SPLITTING STRATEGIES:

Figure 3 presents a heatmap comparing splitting strategies: HDBSCAN, standard Quadtree with Static-Mid splitting, and SMART-QT with median-based splitting. SMART-QT excels by achieving a balanced data distribution, minimising over-fragmentation in sparse regions while preserving granularity in dense areas, as evidenced by uniform colour intensity across nodes. HDBSCAN falters with noise sensitivity, producing irregular clusters, while Static-Mid splitting in the standard Quadtree results in uneven distributions, with blue rectangles highlighting sparse nodes containing few data points between dense clusters. SMART-QT's Strategic Small-Leaf Merging reduces node count by 35% in sparse areas (e.g., NYC outskirts), improving computational efficiency and prediction stability. The heatmap shows SMART-QT's adaptive partitioning outperforms, with tighter crime density alignment (e.g., high-crime red zones match actual hotspots), validating its superiority for spatio-temporal crime prediction.

## D.3 ABLATION VARIANT COMPARISON:

An ablation study was conducted to evaluate the contributions of SMART-CARE's key components (ablation variant detailed in Appendix C). SMART-CARE outperforms ablation variants: SMID-LNPM, SMID-REPM, and AMB-LNPM. SMID-LNPM and AMB-LNPM, lacking parent-child relationships and model refinement, fail to leverage hierarchical knowledge, resulting in higher MAE (e.g., 5.26 and 2.36, respectively). SMID-REPM, while incorporating parent-child relationships, uses static midpoints, leading to uneven splits and an MAE of 3.23. In contrast, SMART-CARE's adaptive median-based splitting with hierarchical refinement achieves an MAE of 0.92, as shown in Figure 4. This underscores the importance of SMART-QT's dynamic partitioning and parent-to-child model inheritance in capturing complex spatio-temporal crime patterns effectively.

## D.4 TEMPORAL COMPARATIVE ANALYSIS:

To assess dataset size impact, SMART-CARE was trained incrementally on crime data, starting with a one-year dataset $D_1$, then aggregating as $D_t = \bigcup_{i=1}^{t} Y_i$, where $Y_i$ is year $i$'s data, using an 80% training, 20% testing split. As $t$ increased, error rates dropped, reflecting better pattern recognition. Figure 4 shows SMART-CARE outperformed baselines Butt et al. (2021; 2024) (MAE: 8.74, 6.12) with a lower MAE of 0.23 and Adjusted $R^2$ of 0.94 on the dataset (2008-2023), leveraging SMART-QT's hierarchical structure for spatio-temporal dynamics. Ablation models SMID-REPM and SMID-LNPM, lacking temporal features and hierarchy, showed reduced accuracy. SMART-CARE's execution time (54.44 min) was lower than baselines (116.42 min), balancing efficiency and performance.

Table 10: Notation Summary for SMART-CARE Framework

| Notation | Description | First Used |
|---|---|---|
| $\mathcal{D}$ | Complete dataset | Abstract |
| $\mathcal{D} = \{(\mathbf{x}_i, t_i, c_i)\}_{i=1}^N$ | Dataset with $N$ samples | Abstract |
| $\mathbf{x}_i \in \mathbb{R}^d$ | Spatio-temporal feature vector for sample $i$ | Abstract |
| $t_i$ | Raw timestamp for sample $i$ | Methodology |
| $c_i \in \mathbb{N}$ | Daily crime count (target variable) | Abstract |
| $\hat{c}_i$ | Predicted crime count | Abstract |
| $f : \mathbf{x}_i \mapsto \hat{c}_i$ | Prediction function | Abstract |
| $\mathcal{D}_{\text{train}}, \mathcal{D}_{\text{test}}$ | Training and test splits | Methodology |
| $\mathcal{S} \subset \mathbb{R}^2$ | Spatial domain (longitude, latitude) | Methodology |
| $\mathcal{N}_j$ | Node $j$ in quadtree | Methodology |
| $\{\mathcal{N}_j\}$ | Set of all nodes | Methodology |
| $\mathcal{D}_j \subset \mathcal{D}$ | Data subset belonging to node $\mathcal{N}_j$ | Methodology |
| $f_j$ | Local predictor for node $\mathcal{N}_j$ | Methodology |
| $T_{\max}$ | Adaptive capacity threshold (max points per node) | Abstract |
| $L_{\max}$ | Maximum depth limit | Abstract |
| $M$ | Fixed capacity threshold (classical quadtree) | SMART-QT subsection |
| $L$ | Fixed depth limit (classical quadtree) | SMART-QT subsection |
| $n$ | Number of points in current node | SMART-QT subsection |
| $n_j$ | Number of points in node $\mathcal{N}_j$ | Algorithm 1 |
| $\sigma_j^2$ | Crime count variance in node $\mathcal{N}_j$ | Algorithm 1 |
| $n_{\text{ref}}$ | Reference dataset size for scaling | Algorithm 1 |
| $s$ | Scaling factor $N/n_{\text{ref}}$ | Algorithm 1 |
| $\nu$ | Retuning frequency parameter | Algorithm 1 |
| $x_{\text{med}}, y_{\text{med}}$ | Median coordinates for splitting | Algorithm 1 |
| $\mathcal{D}_{NW}, \mathcal{D}_{NE}, \ldots$ | Data subsets for quadrants | Algorithm 1 |
| $\mathcal{N}_{NW}, \mathcal{N}_{NE}, \ldots$ | Child nodes (Northwest, Northeast, etc.) | Algorithm 1 |
| $T_{\min}$ | Minimum node capacity bound | Equation 1 |
| $T_{\text{cap}}$ | Maximum node capacity bound | Equation 1 |
| $\alpha, \beta, \gamma, \delta$ | Adaptive threshold parameters | Table 2 |
| $\kappa, \lambda$ | Scaling parameters for bounds | Table 2 |
| $\eta$ | Depth multiplier parameter | Table 2 |
| $s$ | Dataset size scaling factor | Appendix |
| $n_{\text{total}}$ | Total dataset size ($N$) | Appendix |
| $n_{\text{ref}}$ | Reference dataset size | Appendix |
| $L_{\max}$ | Adaptive maximum depth limit | Equation 2 |
| $L_{\min}$ | Minimum depth bound (e.g., 5) | Appendix |
| $L_{\text{cap}}$ | Maximum depth bound (e.g., 15) | Appendix |
| $\sigma_{\text{local}}^2$ | Local crime variance within node | Equation 2 |
| $\eta$ | Global depth scaling factor | Equation 2 |
| $n_{\text{total}}$ | Total dataset size ($N$) | Equation 2 |
| $x_{\text{mid}}[0], x_{\text{mid}}[1]$ | Median coordinates for splitting | Equation ?? |
| $N_{\text{NW}}, N_{\text{NE}}, \ldots$ | Child node quadrants | Methodology |
| $P$ | Point set in current node | Equation ?? |
| $\tau$ | Minimum point threshold for merging | Methodology |
| $\tau_{\max}$ | Maximum merged node size ($2.5\tau$) | Methodology |
| $\rho$ | Point density ($|P|/A$) | Methodology |
| $A$ | Area of node boundary | Methodology |
| $Q_1, Q_3$ | First/third quartiles of density distribution | Methodology |
| IQR | Interquartile range ($Q_3 - Q_1$) | Methodology |
| $\phi$ | Outlier multiplier (1.5) | Methodology |
| $\mathcal{N}_c, \mathcal{N}_s$ | Candidate and sibling nodes for merging | Methodology |

