# OpenReview forum: "Crime Prediction using Adaptive Quadtrees"
_ICLR.cc/2026/Conference — Submitted to ICLR 2026_

### Official Review · Reviewer_9B6w · 2025-11-01

**Soundness:** 2
**Presentation:** 2
**Contribution:** 2
**Rating:** 2
**Confidence:** 4

**Summary:**

The paper proposes a dynamic partitioning algorithm of urban spaces for scalable crime prediction. It utilizes an adaptive quadtree framework with node/capacity thresholds derived from available data, performs spatial balancing through median-based splitting on longitudes/latitudes, retains parent point data for knowledge transfer by referencing from child nodes, and employs density-based small-leaf merging to prevent partitioned regions from being too fine. Additionally, it uses a hierarchical regression scheme for node-level predictions, instantiated with both tree and neural architectures.

**Strengths:**

The paper focuses on spatial partitioning of complex geometries for crime prediction.

It is well-written with an appropriate level of detail.

The results as compared to other works in crime prediction using spatial partitions are promising, noting an improved ability to parse complex relationships in large datasets like NYC and Chicago.

**Weaknesses:**

The proposed approach is a bit dated, focusing on the spatial dimension. Spatial partitioning has been studied well before.

The work doesn't address current SOTA in spatiotemporal forecasting, which focus more on neural approaches. Contemporary methods also model spatiotemporal crime dynamics using probabilistic models such as self-exciting point processes for more accurate forecasting.

The connection to ICLR in terms of insights into learning representations is not clear.

**Questions:**

Can you add comparison to more contemporary spatiotemporal works?

---

### Official Review · Reviewer_WzmL · 2025-11-01

**Soundness:** 3
**Presentation:** 3
**Contribution:** 2
**Rating:** 4
**Confidence:** 2

**Summary:**

This paper introduces SMART-CARE, a novel framework for urban crime prediction that combines an adaptive quadtree (SMART-QT) for spatial partitioning and a hierarchical prediction mechanism (CARE). The quadtree adaptively partitions space based on crime variance and data density, while the CARE ensemble refines predictions at each level using parent-child knowledge transfer. Evaluated on large-scale NYC and Chicago datasets, SMART-CARE outperforms prior methods including clustering, static trees, and deep learning-based transfer learning, with superior accuracy (MAE = 0.92 average) and efficiency. The framework supports both tree-based and neural instantiations, demonstrating flexibility and scalability.

**Strengths:**

* Proposes a well-structured and scalable spatio-temporal prediction framework (SMART-CARE) combining adaptive quadtree partitioning with hierarchical model refinement.
* Effectively utilizes feature propagation and model inheritance to transfer knowledge across spatial scales, enabling fine-grained yet stable local predictions.
* Demonstrates consistent and strong empirical performance on two large-scale, real-world datasets, outperforming several baselines including SARIMA, BiLSTM, static quadtrees, and clustering methods.
* Offers architectural flexibility, supporting both tree-based models (e.g., XGBoost) and neural networks (e.g., GRU, LSTM, Transformer), indicating generalizability.
* Provides thorough implementation details and visualizations to support reproducibility and interpretability.

**Weaknesses:**

* The model inheritance mechanism plays a central role in the proposed framework but lacks an explicit ablation study isolating its effect. Current comparisons (e.g., with AMB-LNPM) involve simultaneous changes in spatial partitioning, making it difficult to assess the independent contribution of inheritance.
* Neural network variants are only lightly analyzed. While the framework claims support for NN models, most discussions and evaluations focus on tree-based models. More detailed analysis on how neural models behave (e.g., with respect to sequence length, stability, hierarchy depth) is missing.
* Most figures and tables use very small fonts, which affects readability and weakens the overall presentation.
* More importantly, the paper’s contribution is primarily in the design of an application-specific forecasting pipeline, with limited connection to representation learning questions. The lack of discussion or analysis around representation learning makes the relevance to ICLR questionable.

**Questions:**

* Have you conducted ablation studies that isolate the effect of model inheritance while keeping the spatial partitioning fixed? This would help clarify how much of the performance gain is specifically attributable to inheritance.
* What role do neural models (GRU, LSTM, Transformer) play within SMART-CARE beyond demonstrating generality? Are there scenarios where they outperform tree-based models or provide unique advantages? Further discussion on performance trade-offs, convergence behavior, and architectural compatibility would be valuable.
* Do you consider any aspect of your method as contributing to general representation learning, or transferable feature learning, beyond the specific domain of crime prediction?

**Details Of Ethics Concerns:**

Since the proposed model is applied to crime prediction, there may be risks of reinforcing societal bias or potential misuse, such as discriminatory targeting or misinterpretation of predictions.

---

### Official Review · Reviewer_dAMW · 2025-11-02

**Soundness:** 1
**Presentation:** 1
**Contribution:** 1
**Rating:** 0
**Confidence:** 5

**Summary:**

I voted for desk reject; see the extensive private discussion among reviewers prior to review.

**Strengths:**

see above

**Weaknesses:**

* Methodologically, the proposed algorithm is not a representation learning approach but rather a partitioning algorithm that offers limited novelty and is largely consistent with existing methods. IMHO: The fit with ICLR is weak, as the paper does not engage with or advance core topics in representation learning.
* The code provided is not reproducible: key components such as hyperparameter tuning procedures are missing / not reported, and several data files are absent. There is also no README nor documentation, making replication impossible.
* Moreover, the paper fails to situate its work within relevant literature. For example, in the domain of crime forecasting, there are extensive theoretical frameworks on spatio-temporal crime dynamics, including self-exciting processes and related models. (The spatial dimension is treated via the partioning but SOTA method often use convolutional / ResNet layers that keep the spatial structure in tact and then predict ahead-of-time) None of these are cited or discussed.
* The empirical evaluation is also insufficient: the baselines and benchmarks (see previous points) are largely missing.
* The benchmarking setup is not realistic (see appendix D.4). It's unclear whether the temporal order is intact; but I think it could predict "previous crime" from "feature "events", leading to information leakage over time.

Overall, the submission lacks novelty, rigor, and relevance, and does not meet the minimum standards for ICLR. I therefore recommend desk rejection to save everyone unnecessary workload.

**Questions:**

see above

---

### Official Review · Reviewer_nTGr · 2025-11-04

**Soundness:** 2
**Presentation:** 2
**Contribution:** 2
**Rating:** 2
**Confidence:** 4

**Summary:**

The paper proposes SMART-CARE, combining an adaptive quadtree for variance-aware median splitting with a hierarchical predictor that propagates parent predictions as features and warm-starts child models. Experiments on NYC and Chicago datasets show large gains, along with O (log n) inference via at-most-one model per tree level.

**Strengths:**

- Clear idea of combining adaptive spatial partitioning with hierarchical residual refinement. The design aims for O(log n) routing at inference.
- Practical top-down training and inference procedure with clear pseudocode; parent-to-child feature propagation and parameter inheritance are easy to implement.
- Evaluated on public datasets and conducted some efficiency discussion;

**Weaknesses:**

- Clarity and writing quality are poor: frequent typos, duplications and inconsistent naming (e.g., line 41“Butt et al. (2021)”; “QREPM” appears once without definition; “AMB_LNPM/SMID-LNPM” introduced but not clearly motivated; “Appendex” spelling). These issues make the paper difficult to follow and cast doubt on rigor.
- Inconsistencies in data description: NYC size is variously “7.8M” and “7.9M”.
- Method novelty is incremental: adaptive quadtrees and warm-starting models are known ideas; the paper mainly aggregates heuristics (variance-aware thresholds, IQR-merging, parent-prediction feature). The contribution is more engineering than a new learning principle, but it’s framed as a major algorithmic advance.
- Baselines are weak, stronger recent spatio-temporal baselines (e.g., GNN, transformer and Mamba) are missing.
- The claimed O (log n) inference depends on enforcing a max depth L_max, not on a proven property of the adaptive quadtree, under skewed spatial distributions there’s no guarantee depth is logarithmic, so the complexity claim isn’t theoretically secured. And the empirical timing results omit key methodological details such as hardware specifications, batch.
- Evaluation protocol is under-specified and risks leakage: the paper uses generic 80/20 split, but for spatiotemporal forecasting one expects time-ordered splits and region-held-out tests. The method heavily uses parent predictions as features; with random splits or standardization done on node-level data that inadvertently includes test points, leakage is plausible. The “D_t = ⋃_{i=1}^t Y_i” description does not clearly state how test years are held out and how lagged features are computed strictly from past data only.
- Feature-importance results raise concerns: “Prediction” (the inherited, model-generated feature) dominates importance, paired with unclear data splits, which could be symptomatic of leakage or target-driven shortcuts rather than robust spatiotemporal reasoning.
- Reproducibility: The paper gives insufficient detail on data processing.

**Questions:**

- How exactly are split thresholds chosen at each node? What are the precise stopping rules beyond minimum leaf size, and are they validated per node or globally?
- What is the principled criterion for merging? Is merging applied once after the tree is grown or iteratively during growth? Does merging depend on label distribution or only counts?
- What is the exact normalization behind MAE ≈ 0.92 / 0.23? Are errors per-leaf, per-day, or normalized by region size?
- Why are strong spatiotemporal MLP, GNN, Transformer and Mamba baselines absent?
- When you standardize the parent predictions “within Dν,” are the statistics computed only on the training subset of that node? How is this handled for test points routed to the same node?

---

### Meta-Review · Area_Chair_iBvm · 2026-01-05

**Summary:**

After careful consideration of all reviews, I recommend rejection. The reviewers reached strong consensus on several fundamental issues. Most critically, three reviewers explicitly noted that this work does not fit ICLR's focus on representation learning, as the proposed method is essentially a spatial partitioning algorithm that aggregates known techniques rather than introducing new learning principles. The methodological contribution is incremental, combining adaptive quadtrees, warm-starting, and variance-based splitting without advancing our theoretical understanding. The experimental evaluation raises serious red flags. The baselines are outdated, conspicuously missing contemporary spatiotemporal methods like GNNs, Transformers, and Mamba-based models that represent current state of the art. More concerning, multiple reviewers identified potential data leakage issues stemming from unclear train-test splits and the suspicious dominance of the parent prediction feature in importance rankings, which could indicate the model is exploiting shortcuts rather than learning robust spatiotemporal patterns. The temporal evaluation protocol remains underspecified, with no clear indication of proper time-ordered holdout or region-based cross-validation. Beyond these substantive issues, the paper suffers from poor presentation quality with numerous typos, inconsistent terminology, and incomplete reproducibility details. While Reviewer WzmL acknowledged the empirical effort, even they questioned the relevance to ICLR. I suggest the authors fundamentally reconsider the framing, conduct rigorous ablations to isolate the contribution of each component, implement proper temporal validation protocols to rule out leakage, and compare against strong modern baselines before resubmitting to a more appropriate venue focused on spatiotemporal data mining.

**Reviewer Concerns:**

No author rebuttal was submitted for this paper, leaving all major reviewer concerns unaddressed.

**Reviewer Scores:**

No author rebuttal was submitted for this paper, leaving no review score changed.

---

### Decision · Program_Chairs · 2026-01-26

Reject